# Multi-scale Temporal Prediction via Incremental Generation and Multi-agent Collaboration

**Zhitao Zeng**[1][†]    **Guojian Yuan**[1][†]    **Junyuan Mao**[1][†]    **Yuxuan Wang**[2]    **Xiaoshuang Jia**[3]

**Yueming Jin**[1][*]

[1]National University of Singapore    [2]Alibaba Group    [3]Renmin University of China

## Abstract

Accurate temporal prediction is the bridge between comprehensive scene understanding and embodied artificial intelligence. However, predicting multiple fine-grained states of scene at multiple temporal scales is difficult for vision-language models. We formalize the Multi-Scale Temporal Prediction (MSTP) task in general and surgical scene by decomposing multi-scale into two orthogonal dimensions: the temporal scale, forecasting states of human and surgery at varying look-ahead intervals, and the state scale, modeling a hierarchy of states in general and surgical scene. For instance in general scene, states of contacting relationship are finer-grained than states of spatial relationship. For instance in surgical scene, medium-level steps are finer-grained than high-level phases yet remain constrained by their encompassing phase. To support this unified task, we introduce the first MSTP Benchmark, featuring synchronized annotations across multiple state scales and temporal scales. We further propose a novel method, Incremental Generation and Multi-agent Collaboration (IG-MC), which integrates two key innovations. Firstly, we propose an plug-and-play incremental generation to keep high-quality temporal prediction that continuously synthesizes up-to-date visual previews at expanding temporal scales to inform multiple decision-making agents, ensuring decision content and generated visuals remain synchronized and preventing performance degradation as look-ahead intervals lengthen. Secondly, we propose a decision-driven multi-agent collaboration framework for multiple states prediction, comprising generation, initiation, and multi-state assessment agents that dynamically triggers and evaluates prediction cycles to balance global coherence and local fidelity. Extensive experiments on the MSTP Benchmark in general and surgical scene show that IG-MC is a generalizable plug-and-play method for MSTP, demonstrating the effectiveness of incremental generation and the stability of decision-driven multi-agent collaboration. The code, model weights, benchmark can be found in https://github.com/jinlab-imvr/MSTP.

## 1  Introduction

The evolution of embodied AI has ushered in systems capable of interpreting human behavior, predicting intentions, and executing physical actions in dynamic general and surgical environments [1, 2, 3]. From assistive robots in healthcare [4, 5] to autonomous agents in smart home settings [6, 7] to surgical robot with agentic systems in operating room, these systems aim to bridge the gap between cognitive reasoning and real-world interaction in healthcare. Traditional robotics, confined to structured tasks with predefined routines, has been revolutionized by models trained

---

[*]Corresponding Author    [†] Equal Contribution

39th Conference on Neural Information Processing Systems (NeurIPS 2025).

on vast multimodal data [8, 9]. These models, such as Large Language Models (LLMs) [10, 11], Multi-Agent Systems (MAS) [12, 13, 14] and Vision-Language models (VLMs) [15, 16, 17, 18], offer unprecedented capabilities in parsing instructions, understanding scenes, and generating action sequences in general scene. However, a critical challenge of embodied learning in surgical scene remains: achieving reliable and trustworthy multi-scale temporal prediction that ensures both short-term action accuracy of healthcare robot and long-term task coherence of surgical embodied agents.

In a parallel vein, recent advances in artificial intelligence [19, 20, 21, 22] have significantly improved analysis and prediction of healthcare robot, achieving promising performance in tasks such as phase recognition [23] and instrument-tissue interaction detection [24], through convolutional [25, 23], recurrent [26, 27, 28], and transformer-based [29, 30, 31] architectures, as well as video generation through diffusion models [32, 33]. Undoubtedly, these approaches made an initial breakthrough, enabling faithful and effective surgical problem-solving. However, ① current work predominantly focuses on single-scale prediction and analysis, either coarse-grained phases or fixed temporal windows [34, 35, 36, 37]. Despite some improvements, this myopic perspective neglects the hierarchical and time-varying nature of surgical decision-making [38]. Additionally, attributed to the accumulation of inaccurate generations in foundation models like VLM and LLM and the increase in visual tokens caused by inputting past multiple frames, ② there is still significant room for improvement in the performance of existing architectures when dealing with surgical workflow analysis and prediction tasks. Additionally, the increasing input length from streaming input frames poses significant challenges for existing methods in terms of computation and memory, leading to performance degradation with the linear growth of temporal input [39]. Furthermore, the accumulation of errors in generative models hinders the effectiveness of existing methods in intelligent surgical problem solving [40, 41].

Several recent studies have incrementally addressed these limitations, though key challenges remain. Across general scenes, long-video MLLMs and planners relax fixed windows and address streaming: Video-MME reveals persistent hour-scale reasoning gaps [42]; VideoStreaming and LongVLM bound token growth via memory propagation and hierarchical segment tokens [43, 44]; and HM-Diffuser targets extendable long-horizon planning with multiscale diffusion [45]. Nonetheless, many pipelines still rely on sparse keyframes or heavy compression, exhibit unfavorable memory/latency scaling, and accumulate rollout errors over long horizons. In surgical scenes, GraSP offers multi-granular supervision unifying pixel-level instruments/actions with step/phase labels [46], while SWAG anticipates minute-scale workflows with single-pass and autoregressive decoders [47]. Yet datasets remain procedure-specific and clip-centric, cross-scale dependencies are only partially modeled, and prospective clinical validation is limited. These gaps motivate an integrated, trustworthy multi-scale temporal predictor that couples short-horizon action accuracy with long-horizon task coherence under realistic streaming constraints.

In light of this, we propose **IG-MC** (Incremental Generation via Multi-agent Collaboration), a unified closed-loop framework for multi-scale temporal prediction. At its core, IG-MC introduces two key innovations: (1) an **incremental generation** mechanism that dynamically synthesizes predicted states and images across expanding temporal scales to maintain state-visual synchronization, thereby reducing error accumulation in long-horizon predictions and ensuring trustworthiness; and (2) a **decision-driven multi-agent collaboration** system comprising specialized VLM-based agents [48, 49] that hierarchically refine predictions from phases to steps while enforcing cross-scale consistency. Departing from approaches that rely on frame histories, IG-MC uses a stateless DM that consumes only the current state–image pair to predict the next state, which then conditions the VG to synthesize the next visual preview [50, 51, 52]; removing temporal buffers reduces memory and speeds up training and inference. Crucially, these innovations are embedded within a closed-loop design, where the state predictions of DM

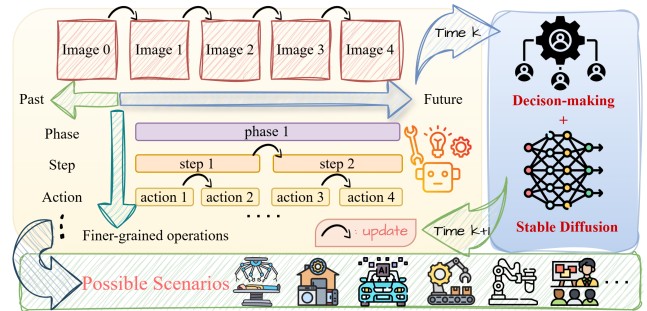

Figure 1: Illustration of the IG-MC overview and advantage across various real-world scenarios.

module and the visual outputs of VG module continuously inform and correct one another. This bidirectional interaction enables real-time error correction, interpretable forecasting, and high-fidelity coherence across both global and local scales.

To rigorously evaluate multi-scale forecasting in general and surgical scene, we introduce the first Multi-scale Temporal Prediction (**MSTP**) benchmark, enhanced with synchronized multi-scale annotations across multiple temporal horizons and hierarchical state. Through extensive experiments, our framework demonstrates excellent performance across various metrics. After adding the plug-and-play modules (DM, VG), most metrics improve significantly.

Our major contributions can be summarized as follow:

- **Incremental Generation** We propose the first incremental generation mechanism that dynamically synthesizes state-image pairs across expanding temporal scales, enabling error-corrected temporal prediction while maintaining state-visual synchronization.
- **Multi-agent Collaboration** We develop a decision-driven multi-agent collaboration framework for multiple fine-grained state prediction, where specialized VLM-based agents hierarchically coordinate temporal predictions while enforcing cross-scale consistency.
- **MSTP Benchmark** We introduce the MSTP benchmark, the first dataset providing synchronized annotations across both temporal horizons and state hierarchies for comprehensive multi-scale evaluation.

## 2   Related Work

**Temporal Prediction** Capturing temporal dynamics at multiple scales is crucial for long-horizon forecasting [53, 54]. Clockwork RNNs partition recurrence across multiple rhythms to learn both fast and slow dynamics [55]. Deep fully-connected stacks such as N-BEATS demonstrate that purely residual-based architectures achieve state-of-the-art results on M3/M4 competitions, offering interpretability and transferability [56]. More recently, self-supervised hierarchical masked modeling methods like HiMTM leverage multi-scale transformers and self-distillation to enhance long-term time series accuracy [57]. Complementary to these, hierarchical learning for long-source generation organizes evidence at coarse-to-fine granularity and improves coherence over extended contexts [58]; multi-token prediction jointly optimizes next-$k$ token heads to stabilize and accelerate training while improving long-horizon fluency [59]; and option-based temporal abstraction provides learnable sub-task policies and termination conditions that align with multi-scale control and forecasting [60].

**Workflow Recognition** Automated recognition of surgical workflow phases and steps has been studied extensively. Early CNN-based multi-task methods [61] such as EndoNet learn phase recognition and instrument presence jointly from laparoscopic videos [25]. SV-RCNet further integrates a ResNet backbone with an LSTM for online workflow segmentation [62]. More recent temporal convolutional approaches [63], notably TeCNO, employ multi-stage dilated causal convolutions to refine phase predictions iteratively and enforce smooth transitions [23]. Building on these, Transformer-based models [64, 65] including Trans-SVNet[66], which uses a hybrid embedding aggregation strategy to fuse spatial and temporal cues, and MuST[37], which introduces multi-scale temporal encoding to capture short-, mid-, and long-term dependencies, further improve accuracy and temporal consistency on standard surgical video benchmarks.

**Visual Generation** Text to image diffusion achieves high fidelity and controllable synthesis through latent diffusion [67] and language conditioned models [68]. Spatial controls based on depth, edge, or segmentation provide precise guidance [69]. In surgical settings, endoscopic frame synthesis based on Stable Diffusion [70] and guided generators for segmentation and triplet constraints [71, 72] produce realistic frames that support augmentation and downstream analysis. Building on these image priors, *video* generation extends control over time. Open domain systems such as Sora improve long range temporal consistency [73]. Surgical variants including Endora [32], SurGen [74], and SurgSora [75] provide controllable previews aligned with surgical workflows for endoscopy and operative scenes.

## 3   Methodology

In this section, we systematically introduce our framework IG-MC (See Fig 2). Initially, we elucidate the methods about incremental generation for multi-scale temporal consistency (Sec 3.2). Subse-

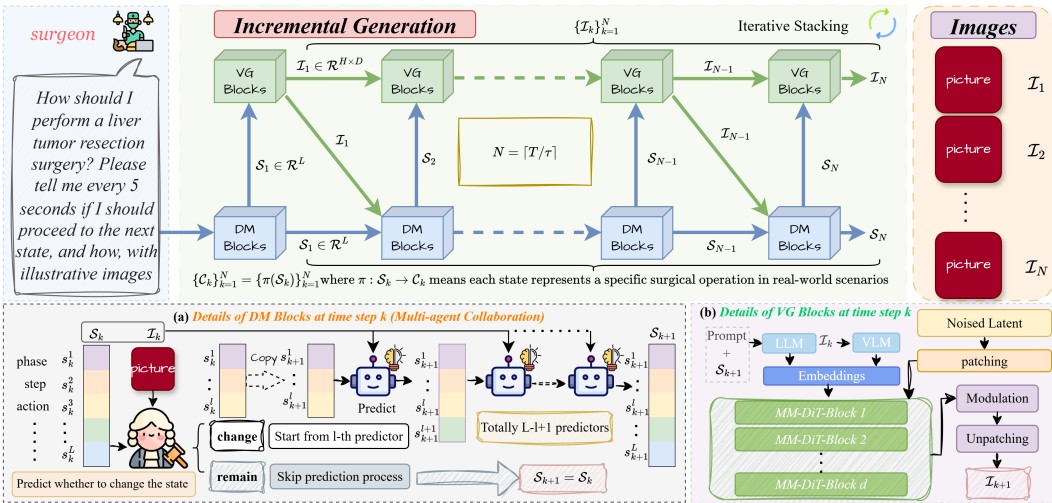

Figure 2: The upper half of the image presents an overview of the model, where Fig (a) and (b) respectively showcase the details of the DM module (Multi-agent Collaboration) and VG module.

quently, we further engage in decision-driven multi-agent collaboration for hierarchical status updates (Sec 3.3). Below, we will present the preliminaries and elaborate on the contributions of each part towards achieving model scalability and SOTA performances.

## 3.1 Preliminaries

**Temporal Scale** We begin by formalizing the temporal prediction problem. Consider a temporal event with total duration $T$ divided into discrete time steps with the temporal scale time interval $\hat{\tau}$, yielding $\hat{N} = \lceil T/\hat{\tau} \rceil$ output points. To improve the effects of predictions, we further consider incremental scale time interval, $\tau$, which is a divisor of $\hat{\tau}$. So there are $N = \lceil T/\tau \rceil$ predicting time points. However, the results will only be presented when the time reaches an output point. At each predicting time point $t$, the system must simultaneously predict both the future state $\mathbf{s}_{t+\tau}$ and generate corresponding visual guidance $\mathcal{I}_{t+\tau}$ for multiple prediction horizons $t \in \{t_1, \ldots, t_N\}$.

**State Scale** The state space $\mathcal{S}$ exhibits a hierarchical structure with multiple states. At the coarsest level, $S_t^1$ captures high-level phases such as "Preparation", while finer levels $S_t^2$ through $S_t^L$ describe progressively more granular steps and actions. This hierarchy induces a natural dependency between levels, where fine-grained steps are semantically constrained by their encompassing phases. Formally, we represent the complete state as $\mathbf{s}_t = (s_t^1, \ldots, s_t^L) \in \mathcal{S}_1 \times \cdots \times \mathcal{S}_L$, where $s_t^\ell$ denotes the state at level $\ell$ and time $t$. Each discrete time point $t_k$ corresponds to a specific operation with environments $C_k \in \mathcal{C}$, where $\mathcal{C} = \{C_1, ..., C_m\}$ represents the complete set of m possible actions.

## 3.2 Incremental Generation

Our incremental generation framework operates through an alternating prediction process that interleaves state forecasting and visual guidance synthesis. At each time step $t_k$, the system first predicts the subsequent state $\mathcal{S}_{k+1}$ by jointly considering both the current state $\mathcal{S}_k$ and the corresponding visual guidance $\mathcal{I}_k$. This decision-making process is formally expressed as:

$$\mathcal{S}_{k+1} = \text{DM}(\mathcal{S}_k, \mathcal{I}_k; \theta_{\text{DM}}) \tag{1}$$

where DM represents our decision-making module (Sec 3.3) with learnable parameters $\theta_{\text{DM}}$, which encodes the temporal evolution patterns of workflows while maintaining consistency with images.

Following state prediction, the system generates the corresponding visual guidance $\mathcal{I}_{k+1}$ for time $t_{k+1}$ through a conditioned diffusion process:

$$\mathcal{I}_{k+1} = \text{VG}(\mathcal{S}_{k+1}, \mathcal{I}_k; \theta_{\text{VG}}) \tag{2}$$

where VG denotes our adapted Stable Diffusion module with parameters $\theta_{\text{VG}}$, specifically optimized for corresponding scenarios. This alternating update scheme creates a tight feedback loop between state prediction and visual synthesis, ensuring that: (1) state predictions remain grounded in visual evidence, and (2) generated guidance images accurately reflect the anticipated progress. The iterative nature of this process allows for error correction across time steps, as both state and visual representations are continuously refined based on each other's outputs.

For a certain procedure of duration $T$ with discrete time intervals $\tau$, the complete sequence of predicted states $\{\mathcal{S}_k\}_{k=1}^N$ and guidance images $\{\mathcal{I}_k\}_{k=1}^N$ (where $N = \lceil T/\tau \rceil$) is generated through the following recurrent process:

$$\begin{cases} \mathcal{S}_{k+1} = \text{DM}\big(\mathcal{S}_k, \mathcal{I}_k; \theta_{\text{DM}}\big) & \text{for } k = 0, \ldots, N-1 \\ \mathcal{I}_{k+1} = \text{VG}\big(\mathcal{S}_{k+1}, \mathcal{I}_k; \theta_{\text{VG}}\big) & \text{for } k = 0, \ldots, N-1 \end{cases} \tag{3}$$

with initial conditions $\mathcal{S}_0$ and $\mathcal{I}_0$ representing the observed state and image at procedure onset. The complete solution can be expressed as the composition of these operations across all time steps:

$$(\{\mathcal{S}_k\}, \{\mathcal{I}_k\}) = \underbrace{\text{DM} \circ \text{VG} \circ \cdots \circ \text{DM} \circ \text{VG}}_{2N \text{ operations}}(\mathcal{S}_0, \mathcal{I}_0) \tag{4}$$

where $\circ$ denotes function composition.

### 3.3 Multi-agent Collaboration

$\diamond$ **Prediction** The decision-making process for state prediction from $t_k$ to $t_{k+1}$ is implemented through a specialized multi-agent collaboration framework. At the core of this system resides a **State Transition Controller (STC)** agent that determines whether the current acting phase requires advancement. The STC receives the current state $\mathcal{S}_k$ and visual guidance $\mathcal{I}_k$ as input, producing either a continuation signal (maintaining the current state) or identifying the specific hierarchical level $l \in \{1, ..., L\}$ where state transition should initiate.

When state transition is required, the system activates a sequence of $L$ LLM-based **State Prediction** agents, where each agent $v_l$ specializes in predicting transitions at level $l$ of the operation hierarchy. These agents are organized such that $v_1$ handles the coarsest-grained phases, while $v_L$ manages the finest-grained steps. Formally, the agent collection $V = \{v_l | v_l \in \mathcal{M}, 1 \leq l \leq L\}$ forms a workflow sequence where the prediction domain of each $v_l$ corresponds to a specific state subspace $\mathcal{S}_t \subset \mathcal{S}$.

The prediction process proceeds through iterative refinement: the STC agent first identifies the starting level $l$ for state changes, after which $v_l$ generates an initial prediction. This output then propagates through subsequent agents $v_{l+1}$ to $v_L$ in a chain, with each agent refining the prediction using both the previous agent's output and the original visual context $\mathcal{I}_k$. This can be expressed as:

$$\mathcal{S}_{k+1} = \sum_{i=1}^N \big[ s_k^i \cdot \mathbb{I}(i < l) + v_i(s_{k+1}^{i-1}, \mathcal{I}_k) \cdot \mathbb{I}(i \geq l) \big] \cdot e_i \tag{5}$$

where $S_{k+1}$ denotes the complete predicted state vector, which is assembled by combining components from all hierarchical levels. Specifically, each component $s_{k+1}^i$ of $S_{k+1}$ follows different update rules based on the level index $i$ relative to the threshold $l$. For indices $i < l$, the component inherits directly from the previous time step's state $s_k^i$, preserving stability in coarse-grained information. For $i \geq l$, the component is computed via the function $v_i$, which generates fine-grained details using the previously assembled lower-level states $S_{k+1}^{i-1}$ and input information $\mathcal{I}_k$. This hierarchical update mechanism is integrated into a unified vector expression through the summation operator. The indicator functions $\mathbb{I}(i < l)$ and $\mathbb{I}(i \geq l)$ ensure the appropriate update rule is applied for each level, returning 1 when the condition is satisfied and 0 otherwise. $+$ means adding. And the standard basis vector $e_i$ maps scalar values to their corresponding positions in the state vector, enabling seamless assembly of the complete state from hierarchical predictions.

$\diamond$ **Fine-tuning** The State Transition Controller agent requires supervised fine-tuning to accurately identify state transition points. In practice, the workflows exhibit significant temporal sparsity in state

changes—the ratio between total time steps $\lceil T/\tau \rceil$ and the numbers of state transition points $\epsilon$ often exceeds 100:1. To address this severe class imbalance, we implement a targeted data augmentation strategy that synthetically increases the proportion of state transition samples. Specifically, we adjust the original imbalanced ratio $\lceil T/\tau \rceil : \epsilon$ applying a multiplicative factor $\alpha$ to $\epsilon$, effectively rebalancing the data set to a 1:1 ratio during training. This augmentation is achieved through temporal window sampling around actual transition points, where each genuine state transition $\epsilon_i$ generates $\alpha$ synthetic variants by perturbing both the input states $\mathcal{S}_k$ and visual contexts $\mathcal{P}_k$ within clinically plausible bounds. The perturbation space covers $\pm\Delta\tau$ timestamp shifts and $\epsilon$-intensity image modifications, preserving the semantic validity of operation transitions while diversifying the training distribution.

## 3.4 Visual Generation

The visual guidance generation module employs a modified Stable Diffusion architecture (VG) to synthesize medically meaningful previews conditioned on both predicted states and prior visual context, as illustrated as Equation 2. The VG module specifically utilizes a latent diffusion paradigm, first encoding the input state $\mathcal{S}_{k+1}$ into a hierarchical embedding space that aligns with the text encoder's semantic structure.

The denoising process is guided by three key conditioning mechanisms: ① the state embedding provides procedural constraints through cross-attention layers, ② the previous guidance image $\mathcal{I}_k$ is injected via residual connections to maintain temporal coherence, and ③ a specific latent space projection ensures anatomical plausibility. This tripartite conditioning yields:

$$\epsilon_\theta(z_t, t, \tau(\mathcal{S}_{k+1}), \phi(\mathcal{I}_k)) \to \hat{\epsilon}_0 \tag{6}$$

where $\epsilon_\theta$ represents the denoising network, $z_t$ the latent noisy image at timestep $t$, $\tau(\cdot)$ the state embedding projector, and $\phi(\cdot)$ the image feature extractor. The output $\hat{\epsilon}_0$ denotes the predicted noise used for iterative latent space refinement.

The diffusion process is optimized using a scenarios-domain constrained objective:

$$\mathcal{L}_{\text{VG}} = \mathbb{E}_{z_0, \epsilon, t} \left[ \|\epsilon - \epsilon_\theta(z_t, t, \tau(\mathcal{S}), \phi(\mathcal{I}))\|_2^2 \right] + \lambda R(\mathcal{I}_{\text{out}}) \tag{7}$$

where $R(\cdot)$ represents a regularization term enforcing tool-tissue interaction realism, and $\lambda$ controls its relative weight. This formulation ensures the generated guidance images maintain both procedural accuracy and visual continuity across temporal predictions.

## 3.5 Integrated IG-MC Framework

The complete IG-MC pipeline operates on certain task ensembles $Q$, where each task $q \in Q$ represents a distinct procedure. Our framework features a decoupled architecture where the DM module and VG module undergo separate training phases. This design enables flexible combination during inference while maintaining modularity. For each sampled task $q$, the DM module generates predicted state trajectories $\{\mathcal{S}_k\}_{k=1}^N$ through iterative application of Equation 1, while the VG module independently produces visual guidance $\{\mathcal{I}_k\}_{k=1}^N$ via Equation 2, with $N = \lceil T/\tau \rceil$ time steps.

The learning objective maximizes the average accuracy between predicted and ground-truth actions:

$$\mathcal{L}_{\text{IG-MC}} = \max_{\theta_{\text{DM}}, \theta_{\text{VG}}} \mathbb{E}_{q \sim Q} \left[ \frac{1}{N} \sum_{k=1}^N P(\mathcal{S}_k = \hat{\mathcal{S}}_k) \mathbb{I}(\frac{k}{\hat{\tau}} \in \mathbb{Z}^+) \right] \tag{8}$$

where $\theta_{\text{DM}}$ and $\theta_{\text{VG}}$ collect all trainable parameters of the DM and VG modules respectively, $\hat{\mathcal{S}}_k$ denotes the ground-truth action at time $t_k$ and $\mathbb{Z}^+$ means the set of positive integers. The expectation is approximated via Monte Carlo sampling over the task distribution $Q$. We provide more detailed derivation and explanation in Appendix J. This objective enforces two crucial properties: (1) differentiability through the state-action mapping, and (2) temporal coherence by equally weighting all time steps. The action accuracy serves as a surrogate metric for overall workflow prediction quality, as the actions directly correspond to related meaningful events. Additionally, other similar metrics are mentioned in the experiments.

# 4 Experiments

In this section, we will validate the effectiveness of our proposed integrated structure plugin, IG-MC. We design four research questions to comprehensively evaluate the performance of IG-MC: **RQ1:** Does IG-MC effectively enhance model performance and applicability? **RQ2:** How do plug-and-play DM and VG module positively affect the model? **RQ3:** How does IG-MC perform in various scale scenarios? **RQ4:** Can IG-MC be applied to different models and real-world scenarios? Through these research questions, we aim to validate the effectiveness and advantages of IG-MC in handling various data from multiple perspectives.

## 4.1 Datasets and Evaluation Metrics

**MSTP Benchmark** Our MSTP Benchmark can be divided into two parts. 1. MSTP Benchmark in general scene (MSTP-General) is built on top of the Action Genome (AG) dataset [76], containing comprehensive annotations of scene graph. We extract the multiple state information of object trajectories including attention, spatial, and contacting relationships between human and object. We augment AG with synchronized three states (attention-level, spatial-level, and contacting-level) at three standard temporal (1s, 2s, and 3s) and multiple dynamic temporal scales from 10s to 60s. 2. MSTP Benchmark in Surgery (MSTP-Surgery) is built on top of the GraSP dataset [77], an endoscopic surgical scene understanding corpus for prostatectomy. We augment GraSP with synchronized two state scales (phase-level and step-level annotations) at four temporal scales (1 s, 5 s, 30 s, and 60 s) to support unified, multi-scale temporal prediction.

**State Prediction Metrics.** We evaluate multi-scale state prediction using Accuracy, Precision, Recall, F1 Score, Jaccard (See Appendix H). Furthermore, the prediction metrics of each state and combined states can be utilized to evaluate the stability of models. P indicates predicted phase, S indicates predicted step, and P&S indicates the combination of predicted phase and step.

**Visual Prediction Metrics.** We evaluate visual prediction performance across several complementary dimensions (See Appendix I): (1) pixel-level fidelity, via Peak Signal-to-Noise Ratio (PSNR); (2) structural consistency, via Structural Similarity Index Measure (SSIM) and its multi-scale variant (MS-SSIM); (3) perceptual realism, via Learned Perceptual Image Patch Similarity (LPIPS) and CLIPScore; (4) distributional alignment, via Fréchet Inception Distance (FID) and Kernel Inception Distance (KID); and (5) retrieval-based congruence, via Recall Precision (R-precision).

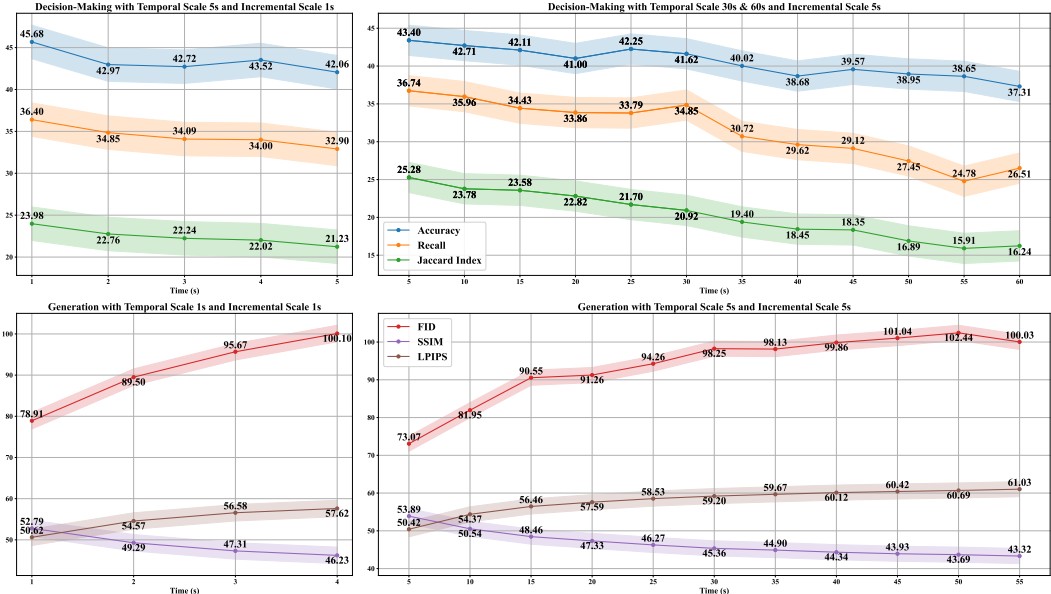

Figure 3: Temporal analysis of proposed IG-MC. The quality of decision-making is decreasing while the quality of generation is also decreasing.

| Model | Temp. Scale | Incr. Scale | State | Accuracy | Precision | Recall | F1 | Jaccard |
|---|---|---|---|---|---|---|---|---|
| **MSTP-General Benchmark** | | | | | | | | |
| Qwen2.5-VL-7B-Instruct[78] | Mixture | – | A&S&C | 13.6 | 17.0 | 13.57 | 12.96 | 7.28 |
| + DM | Mixture | – | A&S&C | 56.11 (+42.51) | 40.29 (+23.29) | 37.39 (+23.82) | 38.57 (+25.61) | 27.14 (+19.86) |
| + DM + VG | Mixture | T/2 | A&S&C | 63.14 (+7.03) | 51.06 (+10.77) | 50.68 (+13.29) | 50.43 (+11.86) | 36.86 (+9.72) |
| Qwen2.5-VL-7B-Instruct[78] | 1–10 | – | A&S&C | 14.17 | 18.06 | 14.81 | 13.47 | 7.61 |
| + DM | 1–10 | – | A&S&C | 58.53 (+44.36) | 40.87 (+22.81) | 39.06 (+24.25) | 39.83 (+26.36) | 28.40 (+20.79) |
| + DM + VG | 1–10 | T/2 | A&S&C | 63.16 (+4.63) | 51.12 (+10.25) | 50.71 (+11.65) | 50.45 (+10.62) | 36.87 (+8.47) |
| Qwen2.5-VL-7B-Instruct[78] | 10–20 | – | A&S&C | 12.17 | 10.32 | 12.47 | 10.24 | 6.21 |
| + DM | 10–20 | – | A&S&C | 52.79 (+40.62) | 39.90 (+29.58) | 34.99 (+22.52) | 36.67 (+26.43) | 25.33 (+19.12) |
| + DM + VG | 10–20 | T/2 | A&S&C | 63.97 (+11.18) | 51.68 (+11.78) | 51.44 (+16.45) | 51.07 (+14.40) | 37.60 (+12.27) |
| Qwen2.5-VL-7B-Instruct[78] | 20–30 | – | A&S&C | 4.76 | 4.26 | 5.43 | 3.88 | 2.71 |
| + DM | 20–30 | – | A&S&C | 51.46 (+46.70) | 46.62 (+42.36) | 41.71 (+36.28) | 43.04 (+39.16) | 29.45 (+26.74) |
| + DM + VG | 20–30 | T/2 | A&S&C | 60.22 (+8.76) | 60.53 (+13.91) | 60.00 (+18.29) | 59.36 (+16.32) | 42.47 (+13.02) |
| Qwen2.5-VL-7B-Instruct[78] | >30 | – | A&S&C | 17.70 | 14.17 | 15.63 | 13.25 | 8.86 |
| + DM | >30 | – | A&S&C | 50.20 (+32.50) | 38.71 (+24.54) | 33.35 (+17.72) | 35.37 (+22.12) | 24.08 (+15.22) |
| + DM + VG | >30 | T/2 | A&S&C | 63.27 (+13.07) | 63.63 (+24.92) | 63.44 (+30.09) | 63.20 (+27.83) | 46.23 (+22.15) |
| **MSTP-Surgery Benchmark** | | | | | | | | |
| Qwen2.5-VL-7B-Instruct[78] | 1 | 1 | P&S | 14.00 | 3.61 | 4.29 | 3.58 | 1.97 |
| + DM | 1 | 1 | P&S | 49.90 (+35.90) | 30.45 (+26.84) | 26.79 (+22.50) | 28.24 (+24.66) | 19.67 (+17.70) |
| Qwen2.5-VL-7B-Instruct[78] | 5 | 5 | P&S | 12.00 | 3.32 | 3.69 | 2.90 | 1.57 |
| + DM | 5 | 5 | P&S | 50.80 (+38.80) | 32.51 (+29.19) | 28.83 (+25.14) | 29.03 (+26.13) | 20.06 (+18.49) |
| + DM + VG | 5 | 1 | P&S | 42.40 (-8.40) | 29.37 (-3.14) | 33.33 (+4.50) | 29.31 (+0.28) | 21.38 (+1.32) |
| Qwen2.5-VL-7B-Instruct[78] | 30 | 30 | P&S | 13.50 | 3.92 | 4.02 | 3.51 | 1.92 |
| + DM | 30 | 30 | P&S | 45.90 (+32.40) | 17.49 (+13.57) | 13.81 (+9.79) | 14.48 (+10.97) | 9.94 (+8.02) |
| + DM + VG | 30 | 5 | P&S | 41.10 (-4.80) | 28.87 (+11.38) | 33.99 (+20.18) | 28.24 (+13.76) | 20.18 (+10.24) |
| Qwen2.5-VL-7B-Instruct[78] | 60 | 60 | P&S | 13.20 | 4.48 | 4.71 | 3.83 | 2.07 |
| + DM | 60 | 60 | P&S | 51.90 (+38.70) | 17.81 (+13.33) | 16.09 (+11.38) | 16.58 (+12.75) | 11.44 (+9.37) |
| + DM + VG | 60 | 5 | P&S | 37.20 (-14.70) | 24.67 (+6.86) | 25.74 (+9.65) | 22.65 (+6.07) | 15.57 (+4.13) |

Table 1: Consolidated comparison of VLMs with plug-and-play DM and VG modules across benchmarks. Bold red (+) indicates improvement and bold blue (–) indicates drop. Temp. Scale indicates temporal scale while Incr. Scale indicates incremental scale; A&S&C are attention, spatial, and contacting relationship, P&S are phase and step.

## 4.2 Evaluating the Efficacy of IG-MC (RQ1 & RQ4)

To address RQ1, we conduct experiments on execution time and various evaluation metrics across various scenarios. As shown in the Figure 3 and Table 1, we can list the **Obs**ervations:

**Obs 1. IG-MC maintains high performance as temporal scales increase.** As shown in Fig 3, we can easily observe that the performance of introducing IG-MC remains at a high level over time. For example, when temporal scale is 5s, the Accuracy remains at 42.06% at time step 5. Although it exhibits a decrease of 3.62% compared to time Step 1, its performance remains favorable.

**Obs 2. IG-MC drives comprehensive improvements across multi-dimensional evaluation metrics.** By analyzing the left side of Table 1, we clearly see that introducing +IG-MC significantly improves model performance on Accuracy, Precision, Recall, F1 and Jaccard metrics. In MSTP-Surgery benchmark, with DM and VG, Accuracy rises from 13.5% → 45.90% , Precision from 3.92% → 17.49%, Recall from 4.02% → 13.81%, etc. This shows the effectiveness of IG-MC in boosting performance in various domains.

**Obs 3. IG-MC generalizes to diverse real-world scenarios, including general human action and surgical workflows.** In general scenarios tested on the Action Genome (AG) dataset, IG-MC achieves an F1 score of 51.07 for joint attention-spatial-contacting state predictions at 10–20s scales, outperforming baselines by 10.24%. In surgical contexts, the framework maintains 28.87% accuracy at 30s temporal scale with 5s incremental updates (24.95% ↑ compared to the baseline). This dual applicability underscores IG-MC's potential as a unified framework for multi-scale prediction in both clinical and general embodied AI scenarios.

### 4.3 Ablation Results (RQ2)

To address RQ2 on the positive impacts of the plug-and-play DM and VG modules, our empirical analysis demonstrates that both components contribute distinct yet complementary improvements to the model's predictive capabilities, with the combined framework outperforming baselines across multi-scale surgical and behavioral forecasting tasks in Table 1.

**Obs 1. The DM module significantly enhances state prediction accuracy and hierarchical consistency.** By integrating surgical workflow knowledge through LLM-based agents, the DM module enables the model to reason about phase-step dependencies and temporal transitions. On the MSTP-Surgery benchmark, adding the DM module to the baseline VLM (Qwen2.5-VL-7B-Instruct) boosts Accuracy from 13.20% to 51.90% at the 60s temporal scale. Notably, this improvement is accompanied by a 13.33-point increase in precision and a 11.38-point increase in recall, indicating that the DM module not only enhances prediction correctness but also reduces false positives and negatives in hierarchical state transitions. For shorter scales such as 5s, the DM module achieves a 38.80-point accuracy gain, highlighting its effectiveness across all temporal horizons.

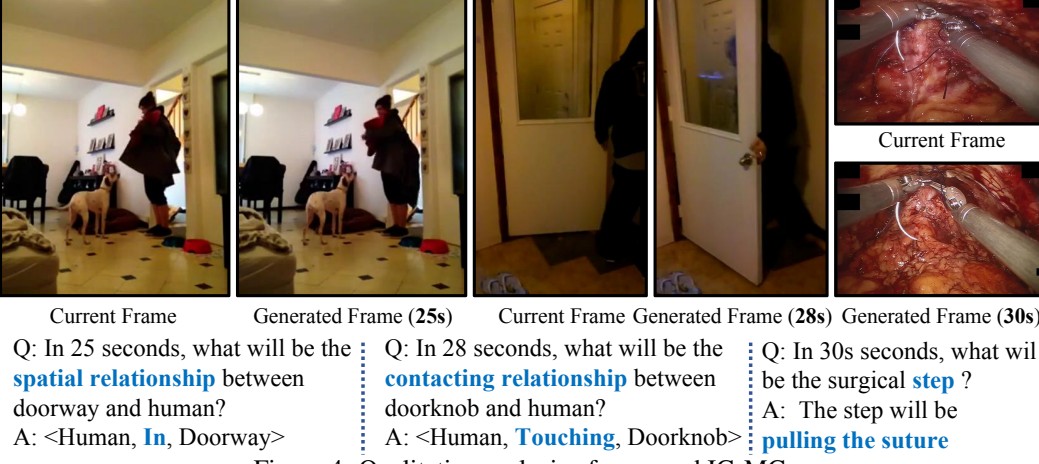

| Current Frame | Generated Frame (**25s**) | Current Frame | Generated Frame (**28s**) | Generated Frame (**30s**) |

Q: In 25 seconds, what will be the **spatial relationship** between doorway and human?
A: <Human, **In**, Doorway>

Q: In 28 seconds, what will be the **contacting relationship** between doorknob and human?
A: <Human, **Touching**, Doorknob>

Q: In 30s seconds, what wil be the surgical **step** ?
A: The step will be **pulling the suture**

Figure 4: Qualitative analysis of proposed IG-MC.

**Obs 2. The VG module further elevates performance by aligning visual guidance with state predictions.** By synthesizing high-fidelity surgical previews conditioned on predicted states, the VG module reinforces temporal coherence and interpretability. In surgical scenarios, at 30s temporal scale, the VG module enhances recall by 11.38% and Jaccard by 9.37% when using a 60s temporal and incremental scale, demonstrating that visual feedback enables more refined step predictions. Crucially, the joint use of DM and VG achieves a 24.73% F1 score improvement over the baseline, underscoring their synergistic role in balancing abstract decision-making with concrete visual grounding.

### 4.4 Performance in Various Scenarios (RQ3)

To address RQ3 on IG-MC's performance across diverse scale scenarios, our comprehensive evaluation demonstrates the framework's exceptional adaptability and robustness across temporal scales, state hierarchies, and incremental time intervals, as illustrated in Table 1 and Fig 4.

**Obs 1. IG-MC excels across varying temporal scales, from short to long horizons.** As shown in Table 1, on the MSTP-Surgery benchmark, IG-MC achieves state prediction accuracies of 50.80% at 5s, 45.90% at 30s, and 51.90% at 60s when using the DM module alone—marking improvements of 38.80%, 32.40%, and 38.70% over baselines, respectively. These results highlight IG-MC's capacity to balance short-term precision and long-term stability, which is a critical advantage for surgical workflow management.

**Obs 2. IG-MC achieves strong performance on hierarchical state scales, from coarse phases to fine steps** On the phase-level state scale of MSTP-Surgery shown in Table 6 (See Appendix E), IG-MC achieves an F1 score of 56.11 when the temporal scale is 5s, compared to the baseline of 13.48. At the finer step-level scale, the framework improves recall by 31.78 points (from 7.94% to 39.72%) at the same temporal scale, showcasing its ability to model granular surgical actions while

respecting hierarchical constraints. In cross-scale consistency tests, the joint phase-step prediction (P&S) achieves a Jaccard index of 20.06 with 5s temporal scale, much higher than the baseline of 1.57, confirming that IG-MC maintains coherence between abstract phases and concrete steps.

**Obs 3. IG-MC adapts effectively to different incremental time scales, optimizing prediction granularity.** When using a 1s incremental scale for 5s temporal predictions in Table 1, IG-MC's DM+VG setup achieves an accuracy of 42.40% on MSTP-Surgery, slightly below the DM-only baseline (50.80%) because of the quality of the pictures generated by VG. But with a 4.50-point recall improvement, it indicates the enhancement of step-level detail of the overall stability. At longer incremental scales (e.g., 5s for 30s temporal predictions), the framework retains 41.10% accuracy while improving recall by 20.18 points, demonstrating that coarser incremental updates can prioritize structural consistency without severe performance degradation. This flexibility allows IG-MC to trade off between computational efficiency and prediction granularity, making it suitable for real-time and planning-oriented scenarios alike.

## 5    Conclusion

In this study, we address the challenge of multi-scale temporal prediction by decomposing the task into temporal and state scales, introducing the MSTP benchmark with synchronized multi-scale annotations for general and surgical scenes. We propose a unified closed-loop framework called IG-MC, which uses an incremental generation mechanism to maintain temporal consistency in state-image synthesis and a decision-driven multi-agent system to hierarchically refine predictions across scales, enabling cross-scale coherence and real-time interaction between state forecasts and visual generation. This work demonstrates a significant advancement in multi-scale prediction accuracy and robustness, offering a promising foundation for enhancing predictions in dynamic scenarios.

## Acknowledgments and Disclosure of Funding

This work was supported by Ministry of Education Tier 2 grant, Singapore (T2EP20224-0028), and Ministry of Education Tier 1 grant, NUS, Singapore (23-0651-P0001).

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

# A   Limitations

While IG-MC demonstrates strong performance across multi-scale prediction tasks, several limitations warrant discussion. (1) The visual prediction quality is contingent on the stability and anatomical accuracy of the Stable Diffusion module. In scenarios with highly specialized surgical tools or rare anatomical variations, VG may generate visually plausible but semantically inconsistent previews, potentially misleading downstream decision agents. (2) The reliance on pre-trained VLMs for base scene understanding implies that its performance is bounded by the VLMs' intrinsic capabilities—for example, limited depth in modeling fine-grained surgical tool interactions in low-resolution imagery. (3) The iterative nature of the incremental generation mechanism introduces non-trivial inference latency, particularly at long temporal scales. Future work could explore lightweight diffusion variants or parallelized agent architectures to address these trade-offs between accuracy and efficiency.

# B   Inferency Latency and Computational Efficiency

We profile computational efficiency and latency on a **single NVIDIA H200** GPU as shown in 2. The end-to-end wall-clock latency is approximately **68 s**. The three decision modules—the State Transition Controller (STC), the phase-level predictor, and the step-level predictor—each take $\sim$20–22 s and together account for $>$90% of total wall-time while operating at only $\approx$1 TFLOPS on average, indicating a *memory-bound* bottleneck rather than a capacity limit. By contrast, the Incremental Generation stage reaches $\approx$97 TFLOPS peak but adds only $\approx$6 s. Peak GPU memory remains modest throughout: $\approx$26 GiB for the decision stack and $\approx$29 GiB for generation.

Table 2: Inference latency and computational efficiency (single NVIDIA H200). "K" denotes $\times 10^3$. Peak memory in GiB.

| Component | Avg. Time (s) | Min (s) | Max (s) | Avg. GFLOPS | Min | Max | Peak GPU Mem (GiB) |
|---|---|---|---|---|---|---|---|
| State Transition Controller | 20.04 | 19.33 | 20.77 | 1.12K | 108.94 | 1.36K | 26.14 |
| Phase Predictor | 20.90 | 19.87 | 21.76 | 1.10K | 109.52 | 1.29K | 26.14 |
| Step Predictor | 21.51 | 20.43 | 22.30 | 1.07K | 90.31 | 1.24K | 26.14 |
| Incremental Generation | 5.81 | 5.78 | 6.10 | 97.32K | 78.62K | 99.71K | 28.53 |

Throughput is not yet real-time; however, the profile suggests clear optimization avenues already underway: *cross-scale weight sharing*, *quantization*, *KV-cache reuse*, and *light MoE pruning*. The numbers below represent an upper bound under the current configuration; with targeted compression and caching, we expect substantial latency reductions toward sub-second responsiveness appropriate for intra-operative scenarios.

# C   Datasets and Evaluation Metrics

**MSTP Benchmark** Our MSTP Benchmark in Surgery is built on top of the GraSP dataset [77], an endoscopic surgical scene understanding corpus for prostatectomy. We augment GraSP with synchronized two state scales (phase-level and step-level annotations) at four temporal scales (1 s, 5 s, 30 s, and 60 s) to support unified, multi-scale temporal prediction.

GraSP comprises annotated videos of 13 cases at 1280×800 resolution and 1 fps; following the official protocol, cases 1, 2, 3, 4, 7, 14, 15, and 21 are used for training, and cases 41, 47, 50, and 51 for testing.

We report dataset sizes exclusively for MSTP, which is constructed by re-sampling the GraSP corpus of robot-assisted prostatectomy videos. From GraSP's 32 h / 13-video source, we generate scale-aware future-prediction at four horizons (1 s, 5 s, 30 s, 60 s) and split them 10:1 into train/test, yielding 40k training and 4k test clips in total. At

Table 3: Dataset analysis of proposed MSTP benchmark.

| Temporal Scale | Train | Test | Details |
|---|---|---|---|
| 1 s | 10k | 1k | 2 frames, 4 states |
| 5 s | 10k | 1k | 6 frames, 12 states |
| 30 s | 10k | 1k | 31 frames, 62 states |
| 60 s | 10k | 1k | 61 frames,122 states |

each scale there are 10k training and 1k test clips (Table 3). Each sample comprises two parts: (1) the *current* image and its states, and (2) the *future* frames and their states. Windows are extracted at the native 30 fps and aligned so the first frame index is shared across scales.

MSTP provides hierarchical supervision with two nested tiers: *Phase* (11 classes) and *Step* (21 classes), where fine-grained *Step* labels are strictly contained within their parent *Phase*. The temporal construction per horizon determines the number of frames and states per clip; e.g., 1 s clips contain 2 frames and 4 states, while 60 s clips

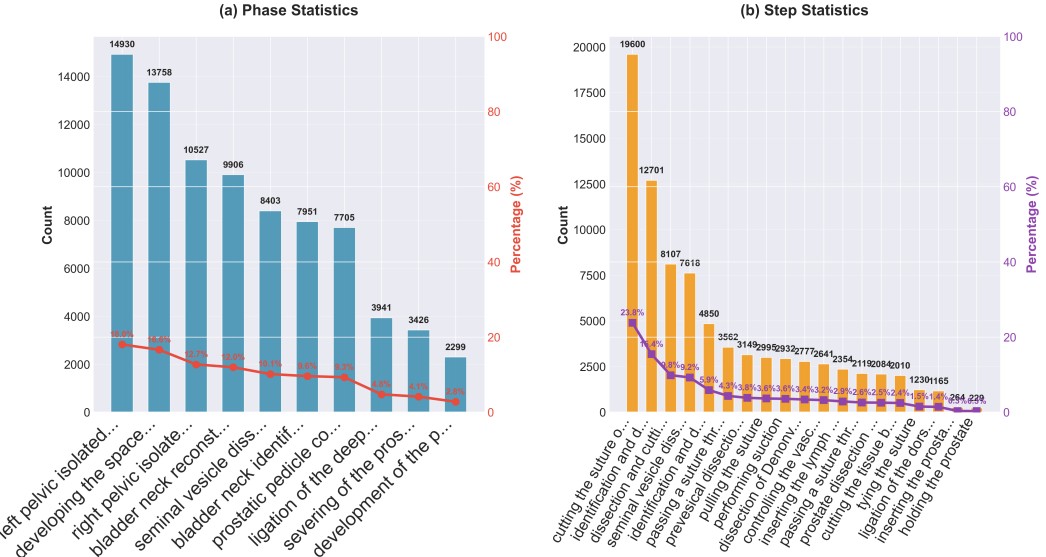

Figure 5: Dataset analysis of proposed MSTP.

contain 61 frames and 122 states. This design ensures temporally consistent inputs across scales and supports coherent hierarchical learning for surgical temporal reasoning.

## D    Implementation Details

**Generation Agent** Our Generation Agent is implemented via the Stable Diffusion 3.5 Large architecture [79], fine-tuned on $4\times$ NVIDIA H100 GPUs in mixed-precision (bf16). Starting from a publicly available pre-trained checkpoint, we trained for one epoch with a per-GPU batch size of 34 (no gradient accumulation). Optimization was performed using AdamW with bf16 precision, a cosine learning-rate schedule and a 10% warm-up phase, weight decay set to 0, gradient checkpointing, and XFormers memory-efficient attention. During training, all images were resized so that their shorter edge measured 1024 px (maintaining aspect ratio) with a minimum resolution of 1024 px. No EMA updates were applied. At inference, we used 30 denoising steps, a CFG scale of 7.5, and the negative prompt "blurry, cropped, ugly."

**Decision-making Agent** The Decision Agent leverages VLMs including Qwen2.5-VL-7B-Instruct [80], fine-tuned on our MSTP decision dataset using $4\times$ NVIDIA H100 GPUs in mixed-precision (bf16) with TF32 support and gradient checkpointing. We employed a per-device batch size of 32, no gradient accumulation, and the AdamW optimizer (initial learning rate $2 \times 10^{-5}$, cosine decay, 10% warm-up, weight decay $1 \times 10^{-2}$). Visual inputs were resized so that the shorter side was 512 px, and text inputs tokenized with the Qwen tokenizer; modality lengths were grouped to align sequence lengths. At inference, greedy decoding was applied (maximum length 128 tokens) with top-$p$ sampling ($p = 0.9$).

## E    Visual Generation Analysis

To isolate the contribution of the VG module, we pair the baseline VLM with VG alone (no multi-agent collaboration) and evaluate continual—rather than hierarchical—state prediction. The VG-only ablation still yields clear gains, demonstrating that VG generalizes across non-hierarchical and hierarchical formulations. In practice, IG, MC, and VG constitute a reproducible, plug-and-play toolkit that reliably improves temporal prediction across diverse VLM backbones.

We provide a dedicated analysis, including (i) decision errors made by different agents and (ii) instances where generated visuals are of insufficient quality, alongside their causes and potential remedies. To quantify accumulated errors in *incremental generation*, we analyze the relationship between decision accuracy and image quality measured by the Fréchet Inception Distance (FID; lower is better) across prediction horizons.

As shown in Table 7, image quality degrades with longer horizons, yet decision accuracy remains comparatively resilient. Empirically, we observe a strong negative correlation between accuracy and FID,

$$\text{FID} \ = \ \alpha \ - \ \beta \times \text{Accuracy}, \quad (R < 0, \ p < 0.05), \tag{9}$$

Table 4: Comparison of General VLMs with plug-and-play DM and VG modules. TS = temporal scale, IS = incremental scale.

| Base | Variant | TS | IS | Accuracy | Precision | Recall |
|------|---------|----|----|----------|-----------|--------|
| LLaVA1.5-7B | Base | 1 | 1 | 13.60 | 3.75 | 3.34 |
| | +DM | 1 | 1 | 30.00(+16.40) | 23.50(+19.75) | 13.80(+10.46) |
| | Base | 5 | 5 | 12.20 | 2.07 | 2.38 |
| | +DM | 5 | 5 | 27.90(+15.70) | 26.03(+23.96) | 14.95(+12.57) |
| | +DM+VG | **5** | **1** | **46.40(+16.40)** | **36.45(+12.95)** | **36.88(+23.08)** |
| | Base | 30 | 30 | 17.00 | 4.17 | 2.77 |
| | +DM | 30 | 30 | 28.00(+11.00) | 21.66(+17.49) | 11.97(+9.20) |
| | +DM+VG | **30** | **5** | **44.20(+16.30)** | **32.92(+11.26)** | **32.04(+20.07)** |
| | Base | 60 | 60 | 18.10 | 4.61 | 3.90 |
| | +DM | 60 | 60 | 29.30(+11.20) | 15.86(+11.25) | 9.60(+5.70) |
| | +DM+VG | **60** | **5** | **38.60(+10.70)** | **26.95(+11.09)** | **28.21(+13.26)** |
| Gemma3-27B | Base | 1 | 1 | 1.80 | 2.66 | 2.75 |
| | +DM | 1 | 1 | 21.00(+19.20) | 6.65(+3.99) | 5.19(+2.44) |
| | Base | 5 | 5 | 2.00 | 1.06 | 1.13 |
| | +DM | 5 | 5 | 19.80(+17.80) | 6.18(+5.12) | 4.90(+3.77) |
| | +DM+VG | **5** | **1** | **34.10(+14.30)** | **19.30(+13.12)** | **20.46(+15.56)** |
| | Base | 30 | 30 | 2.00 | 0.80 | 1.54 |
| | +DM | 30 | 30 | 24.30(+22.30) | 7.15(+6.35) | 5.38(+4.04) |
| | +DM+VG | **30** | **5** | **38.10(+13.80)** | **26.83(+19.68)** | **25.44(+20.06)** |
| | Base | 60 | 60 | 1.60 | 0.88 | 0.52 |
| | +DM | 60 | 60 | 26.90(+25.30) | 8.09(+7.21) | 5.69(+5.17) |
| | +DM+VG | **60** | **5** | **34.60(+7.70)** | **20.76(+12.67)** | **22.76(+17.07)** |
| InternVL3-8B | Base | 1 | 1 | 13.60 | 3.61 | 3.42 |
| | +DM | 1 | 1 | 36.20(+22.60) | 23.22(+19.61) | 17.18(+13.76) |
| | Base | 5 | 5 | 13.80 | 3.48 | 3.93 |
| | +DM | 5 | 5 | 37.30(+23.50) | 25.60(+22.12) | 19.62(+15.69) |
| | +DM+VG | **5** | **1** | **45.60(+8.30)** | **27.01(+1.41)** | **28.53(+8.91)** |
| | Base | 30 | 30 | 14.40 | 2.11 | 3.69 |
| | +DM | 30 | 30 | 42.30(+27.90) | 20.99(+18.88) | 18.04(+14.35) |
| | +DM+VG | **30** | **5** | **40.80(-1.50)** | **25.54(+4.55)** | **28.18(+10.14)** |
| | Base | 60 | 60 | 16.70 | 6.01 | 5.26 |
| | +DM | 60 | 60 | 36.30(+19.60) | 19.29(+13.28) | 14.92(+9.66) |
| | +DM+VG | **60** | **5** | **38.40(+2.10)** | **22.44(+3.15)** | **23.88(+8.96)** |
| Qwen2.5-VL-7B | Base | 1 | 1 | 14.00 | 3.61 | 4.29 |
| | +DM | 1 | 1 | 49.90(+35.90) | 30.45(+26.84) | 26.79(+22.50) |
| | Base | 5 | 5 | 12.00 | 3.32 | 3.69 |
| | +DM | 5 | 5 | 50.80(+38.80) | 32.51(+29.19) | 28.83(+25.14) |
| | +DM+VG | **5** | **1** | **42.40(-8.40)** | **29.37(-3.14)** | **33.33(+4.50)** |
| | Base | 30 | 30 | 13.50 | 3.92 | 4.02 |
| | +DM | 30 | 30 | 45.90(+32.40) | 17.49(+13.57) | 13.81(+9.79) |
| | +DM+VG | **30** | **5** | **41.10(-4.80)** | **28.87(+11.38)** | **33.99(+20.18)** |
| | Base | 60 | 60 | 13.20 | 4.48 | 4.71 |
| | +DM | 60 | 60 | 51.90(+38.70) | 17.81(+13.33) | 16.09(+11.38) |
| | +DM+VG | **60** | **5** | **37.20(-14.70)** | **24.67(+6.86)** | **25.74(+9.65)** |

Table 5: Comparison of **Surgical VLMs** with plug-and-play DM and VG modules. TS = temporal scale, IS = incremental scale.

| Base | Variant | TS | IS | Accuracy | Precision | Recall |
|---|---|---|---|---|---|---|
| | Base | 1 | 1 | 1.20 | 3.73 | 2.85 |
| | +DM | 1 | 1 | 41.90(+40.70) | 3.22(-0.51) | 2.58(-0.27) |
| | Base | 5 | 5 | 1.06 | 4.68 | 2.79 |
| | +DM | 5 | 5 | 42.70(+41.64) | 26.98(+22.30) | 22.91(+20.12) |
| | +DM+VG | **5** | **1** | **44.84(+2.14)** | **28.43(+1.45)** | **29.06(+6.15)** |
| SurgVLM-7B | Base | 30 | 30 | 12.80 | 4.02 | 3.39 |
| | +DM | 30 | 30 | 42.30(+29.50) | 20.97(+16.95) | 18.63(+15.24) |
| | +DM+VG | **30** | **5** | **40.58(-1.72)** | **26.68(+5.71)** | **26.07(+7.44)** |
| | Base | 60 | 60 | 10.90 | 2.99 | 2.98 |
| | +DM | 60 | 60 | 38.50(+27.60) | 17.95(+14.96) | 15.08(+12.10) |
| | +DM+VG | **60** | **5** | **36.24(-2.26)** | **19.63(+1.68)** | **21.32(+6.24)** |

| Model | Temp. Scale | State Scale | Accuracy | Precision | Recall | F1 | Jaccard |
|---|---|---|---|---|---|---|---|
| Qwen2.5-VL-7B | 1 | Phase | 20.50 | 13.17 | 14.41 | 13.04 | 7.42 |
| + DM | 1 | Phase | 73.10 (+52.60) | 47.74 (+34.57) | 44.98 (+30.57) | 46.27 (+33.23) | 37.99 (+30.57) |
| Qwen2.5-VL-7B | 1 | Step | 20.30 | 8.27 | 9.23 | 8.35 | 4.71 |
| + DM | 1 | Step | 50.60 (+30.30) | 42.87 (+34.60) | 39.90 (+30.67) | 41.08 (+32.73) | 28.68 (+23.97) |
| Qwen2.5-VL-7B | 1 | Phase&Step | 14.00 | 3.61 | 4.29 | 3.58 | 1.97 |
| + DM | 1 | Phase&Step | 49.90 (+35.90) | 30.45 (+26.84) | 26.79 (+22.50) | 28.24 (+24.66) | 19.67 (+17.70) |
| Qwen2.5-VL-7B | 5 | Phase | 19.90 | 14.41 | 15.14 | 13.48 | 7.64 |
| + DM | 5 | Phase | 81.50 (+61.60) | 58.21 (+43.80) | 54.80 (+39.66) | 56.11 (+42.63) | 48.77 (+41.13) |
| Qwen2.5-VL-7B | 5 | Step | 16.40 | 7.90 | 7.94 | 7.20 | 3.96 |
| + DM | 5 | Step | 52.00 (+35.60) | 45.08 (+37.18) | 39.72 (+31.78) | 40.85 (+33.65) | 28.54 (+24.58) |
| Qwen2.5-VL-7B | 5 | Phase&Step | 12.00 | 3.32 | 3.69 | 2.90 | 1.57 |
| + DM | 5 | Phase&Step | 50.80 (+38.80) | 32.51 (+29.19) | 28.83 (+25.14) | 29.03 (+26.13) | 20.06 (+18.49) |
| Qwen2.5-VL-7B | 30 | Phase | 17.80 | 14.88 | 12.65 | 12.29 | 6.86 |
| + DM | 30 | Phase | 64.40 (+46.60) | 39.54 (+24.66) | 31.66 (+19.01) | 34.65 (+22.36) | 27.03 (+20.17) |
| Qwen2.5-VL-7B | 30 | Step | 18.20 | 7.62 | 7.99 | 7.09 | 3.93 |
| + DM | 30 | Step | 50.90 (+32.70) | 38.91 (+31.29) | 36.09 (+28.10) | 36.24 (+29.15) | 25.27 (+21.34) |
| Qwen2.5-VL-7B | 30 | Phase&Step | 13.50 | 3.92 | 4.02 | 3.51 | 1.92 |
| + DM | 30 | Phase&Step | 45.90 (+32.40) | 17.49 (+13.57) | 13.81 (+9.79) | 14.48 (+10.97) | 9.94 (+8.02) |
| Qwen2.5-VL-7B | 60 | Phase | 20.30 | 16.05 | 15.22 | 14.47 | 8.04 |
| + DM | 60 | Phase | 70.80 (+50.50) | 40.67 (+24.62) | 35.45 (+20.23) | 37.67 (+23.20) | 30.47 (+22.43) |
| Qwen2.5-VL-7B | 60 | Step | 18.40 | 7.64 | 8.25 | 7.23 | 4.00 |
| + DM | 60 | Step | 54.50 (+36.10) | 36.10 (+28.46) | 34.61 (+26.36) | 35.06 (+27.83) | 24.19 (+20.19) |
| Qwen2.5-VL-7B | 60 | Phase&Step | 13.20 | 4.48 | 4.71 | 3.83 | 2.07 |
| + DM | 60 | Phase&Step | 51.90 (+38.70) | 17.81 (+13.33) | 16.09 (+11.38) | 16.58 (+12.75) | 11.44 (+9.37) |

Table 6: Comparison of Qwen2.5-VL-7B-Instruct with and without plug-and-play DM module on MSTP Benchmark.

| Model | Temporal Scale | Incremental Scale | Accuracy | | Time (s) | Accuracy (%) | FID |
|---|---|---|---|---|---|---|---|
| Gemma3-27B | 5 | 5 | 2.00 | | 5 | 43.30 | 70.63 |
| +VG | 5 | 1 | 26.90(+24.90) | | 10 | 41.34 | 83.12 |
| Gemma3-27B | 30 | 30 | 2.00 | | 15 | 42.44 | 85.62 |
| +VG | 30 | 5 | 28.40(+26.40) | | 20 | 41.24 | 88.57 |
| Gemma3-27B | 60 | 60 | 1.60 | | 25 | 41.58 | 90.99 |
| +VG | 60 | 5 | 25.90(+24.30) | | 30 | 40.58 | 94.82 |

Table 7: **Left:** Effectiveness of the VG module on non-hierarchical prediction. **Right:** Temporal relationship between decision accuracy and image quality (FID; lower is better) across horizons.

where $\alpha$ and $\beta$ are empirically fitted constants. In practice, from a 5 s horizon to 30 s, FID increases by about 24 points (from 70.63 to 94.82; $\approx 34\%$ increase), while accuracy decreases by only 2.7 points (from 43.3% to 40.58%). This suggests that the decision-making stack retains robustness even as generative fidelity declines.

## F   Case Analysis with Generated Frames

In the first set of cases about Bladder Neck Reconstruction in Figure 6, despite a noticeable discrepancy between the predicted frames and ground-truth frames that leads to a degradation in video prediction accuracy, state prediction remains unaffected. This is attributable to the fact that the actions depicted in the frames inherently encode the essential motion patterns—while the visual details of the predicted frames differ from the ground truth, they still effectively capture the core dynamics of the real-world actions from an alternative perspective. Consequently, such visual discrepancies do not propagate as interference to subsequent state prediction tasks.

Conversely, in the second set of cases in Figure 6, at certain timesteps where action states should have remained unchanged, the model erroneously predicts the next sequential action in the frames. This misalignment between frame predictions and the actual static action states result in a concurrent decline in both frame prediction quality and subsequent state prediction performance. A key observation here is that the model exhibits a tendency to prematurely generate the next-step action output rather than maintaining the current state when no meaningful motion change is required.

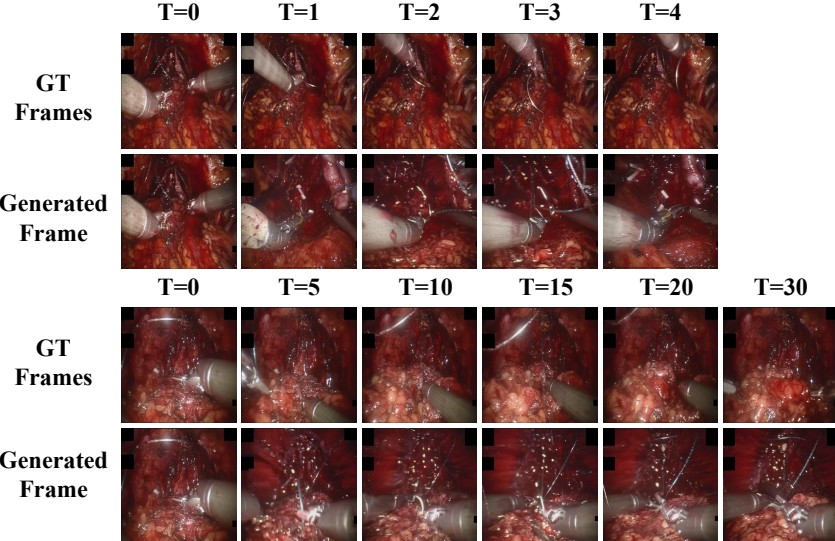

Figure 6: Failure case of generated frames.

## G   Multi-agent Collaboration Analysis

We further investigate the robustness of *multi-agent collaboration* (MC) under cumulative error in Table 8. If the three agents—State Transition Controller (STC), Phase Predictor, and Step Predictor—fail independently, the overall accuracy will follow the product bound $\Pi$ of their marginal accuracies, collapsing near 11–15%. In practice, MC sustains 36–45% across horizons, consistently exceeding $\Pi$ by more than 25 percentage points at every scale. We attribute this to the STC's switch/stay gating mechanism and the shared visual context, which allow downstream agents to correct upstream missteps and prevent multiplicative drift. Notably, the degradation with horizon length remains sub-linear (44.8% $\rightarrow$ 36.2% from 5 s to 60 s), satisfying our in-the-loop clinical threshold of $\geq 35\%$ top-1 accuracy.

## H   State Prediction Metrics

The accuracy metric (*Acc*) represents the proportion of correctly classified frames across the entire video, computed as an overall video-level measure. In contrast, *PR*, *RE*, $F_1$ and *JA* are first determined separately for each phase category before being aggregated through averaging to obtain the final video-level metrics. Letting *Pred* denote the set of predicted frames and *GT* the set of ground truth frames for a particular phase, these metrics are mathematically defined as:

Table 8: Cumulative error analysis of multi-agent collaboration (MC). $\Pi$ denotes the independence/product bound. MC consistently exceeds $\Pi$ by >25 pp.

| Scale | STC Acc. | Phase Acc. | Step Acc. | $\Pi$ | MC Acc. | MC – $\Pi$ |
|-------|----------|------------|-----------|-------|---------|------------|
| 5 s | 57.1 | 51.9 | 49.9 | 14.8 | 44.8 | **+30.0** |
| 30 s | 55.4 | 43.8 | 58.5 | 14.2 | 40.6 | **+26.4** |
| 60 s | 54.9 | 33.2 | 58.8 | 10.7 | 36.2 | **+25.5** |

$$Acc = \frac{\text{Pred}}{\text{GT}} \tag{10}$$

$$PR = \frac{\text{GT} \cap \text{Pred}}{\text{Pred}} \tag{11}$$

$$RE = \frac{\text{GT} \cap \text{Pred}}{\text{GT}} \tag{12}$$

$$JA = \frac{\text{GT} \cap \text{Pred}}{\text{GT} \cup \text{Pred}} \tag{13}$$

$$F_1 = 2 \times \frac{PR \times RE}{PR + RE} \tag{14}$$

# I   Visual Prediction Metrics

We assess visual prediction performance through five complementary dimensions:

**Pixel-level Fidelity:** Peak Signal-to-Noise Ratio (PSNR) measures reconstruction quality:

$$\text{PSNR} = 10 \log_{10} \left( \frac{\text{MAX}_I^2}{\text{MSE}} \right) \tag{15}$$

where $\text{MAX}_I$ is the maximum pixel value (e.g., 255 for 8-bit images) and MSE is the mean squared error.

**Structural Consistency:** Structural Similarity Index (SSIM) evaluates structural preservation:

$$\text{SSIM}(x, y) = \frac{(2\mu_x\mu_y + C_1)(2\sigma_{xy} + C_2)}{(\mu_x^2 + \mu_y^2 + C_1)(\sigma_x^2 + \sigma_y^2 + C_2)} \tag{16}$$

Multi-scale variant (MS-SSIM) extends this across spatial resolutions.

**Perceptual Realism:** Learned Perceptual Image Patch Similarity (LPIPS) uses deep features:

$$\text{LPIPS} = \sum_l \frac{1}{H_l W_l} \sum_{h,w} \|\phi_l(I)_{h,w} - \phi_l(\hat{I})_{h,w}\|_2^2 \tag{17}$$

where $\phi_l$ denotes features from layer $l$ of a pretrained network. CLIPScore measures semantic alignment:

$$\text{CLIPScore} = \max(100 \times \cos(\phi_{\text{CLIP}}(I), \phi_{\text{CLIP}}(T)), 0) \tag{18}$$

with $T$ being text prompts.

**Distributional Alignment:** Fréchet Inception Distance (FID) compares feature distributions:

$$\text{FID} = \|\mu_r - \mu_g\|^2 + \text{Tr}(\Sigma_r + \Sigma_g - 2(\Sigma_r\Sigma_g)^{1/2}) \tag{19}$$

where $(\mu_r, \Sigma_r)$ and $(\mu_g, \Sigma_g)$ are statistics of real/fake features. Kernel Inception Distance (KID) uses polynomial kernels:

$$\text{KID} = \mathbb{E}[k(x_r, x_r') + k(x_g, x_g') - 2k(x_r, x_g)] \tag{20}$$

**Retrieval-based Congruence:** R-precision measures retrieval accuracy:

$$\text{R-precision} = \frac{\text{\# relevant items in top-}R}{\min(R, \text{total relevant})} \tag{21}$$

where $R$ is the number of ground-truth relevant items.

## J Integrated IG-MC Framework

The complete IG-MC pipeline operates on certain task ensembles $Q$, where each task $q \in Q$ represents a distinct procedure. Our framework features a decoupled architecture where the DM module and VG module undergo separate training phases. This design enables flexible combination during inference while maintaining modularity. For each sampled task $q$, the DM module generates predicted state trajectories $\{\mathcal{S}_k\}_{k=1}^N$ through iterative application of the decision-making function:

$$\mathcal{S}_{k+1} = \text{DM}(\mathcal{S}_k, \mathcal{I}_k; \theta_{\text{DM}}), \tag{22}$$

where $\theta_{\text{DM}}$ denotes the trainable parameters of the DM module, $\mathcal{S}_k$ represents the predicted state at time step $t_k$, and $\mathcal{I}_k$ is the visual guidance synthesized up to $t_k$. Concurrently, the VG module produces the visual sequence $\{\mathcal{I}_k\}_{k=1}^N$ through a conditioned diffusion process:

$$\mathcal{I}_{k+1} = \text{VG}(\mathcal{S}_{k+1}, \mathcal{I}_k; \theta_{\text{VG}}), \tag{23}$$

where $\theta_{\text{VG}}$ parameterizes the VG module, and $\mathcal{I}_{k+1}$ is synthesized by denoising a latent representation conditioned on both the predicted state $\mathcal{S}_{k+1}$ and previous visual guidance $\mathcal{I}_k$. The temporal resolution is determined by $N = \lceil T/\tau \rceil$ time steps, with $T$ being the total procedure duration and $\tau$ the incremental time interval.

The learning objective maximizes the temporal average accuracy of state predictions relative to ground-truth annotations:

$$\mathcal{L}_{\text{IG-MC}} = \max_{\theta_{\text{DM}}, \theta_{\text{VG}}} \mathbb{E}_{q \sim Q} \left[ \frac{1}{N} \sum_{k=1}^N P(\mathcal{S}_k = \hat{\mathcal{S}}_k) \mathbb{I} \left( \frac{k}{\hat{\tau}} \in \mathbb{Z}^+ \right) \right], \tag{24}$$

where $\hat{\mathcal{S}}_k$ is the ground-truth state at $t_k$, $\hat{\tau}$ represents the temporal resolution of ground-truth annotations, and $\mathbb{I}(\cdot)$ is an indicator function enforcing temporal alignment. The expectation $\mathbb{E}_{q \sim Q}$ is approximated via Monte Carlo sampling over the task distribution $Q$. The probability term $P(\mathcal{S}_k = \hat{\mathcal{S}}_k)$ derives from a cross-entropy loss between predicted and true state distributions, ensuring differentiability throughout the optimization process.

The term $P(\mathcal{S}_k = \hat{\mathcal{S}}_k)$ represents the probability that the predicted state $\mathcal{S}_k$ matches the ground-truth state $\hat{\mathcal{S}}_k$ at time step $t_k$. This probability serves as a direct measure of prediction accuracy, where higher values indicate better alignment between predicted and actual states.

However, not every incremental time step requires state prediction. The framework operates with two distinct temporal resolutions: the *incremental scale* $\tau$ (e.g., 5s) for internal state updates and the *temporal scale* $\hat{\tau}$ (e.g., 30s) for meaningful prediction outputs. The indicator function $\mathbb{I}(\frac{k}{\hat{\tau}} \in \mathbb{Z}^+)$ enforces this distinction by evaluating to 1 only when the current step index $k$ corresponds to an integer multiple of the prediction interval ratio $\frac{\hat{\tau}}{\tau}$.

Mathematically, this condition:

$$\frac{k}{\hat{\tau}} \in \mathbb{Z}^+ \tag{25}$$

ensures that state predictions are generated precisely at the coarser temporal time steps (every $\hat{\tau}$ seconds), while allowing continuous internal updates at finer incremental intervals. For the example where $\tau = 5s$ and $\hat{\tau} = 30s$, predictions would occur at every 6th incremental step (since $30/5 = 6$), maintaining computational efficiency without sacrificing temporal granularity where needed.

