# OpenReview forum: "Multi-scale Temporal Prediction via Incremental Generation and Multi-agent Collaboration"
_NeurIPS.cc/2025/Conference — NeurIPS 2025 poster_

### Official Review · Reviewer_eUMj · 2025-07-01

**Clarity:** 3
**Significance:** 3
**Originality:** 3
**Rating:** 5
**Confidence:** 3

**Summary:**

The paper presents the IG-MC framework, a novel solution for multi-scale temporal prediction (MSTP) that addresses long-horizon error accumulation and cross-scale consistency through two core innovations: incremental generation and decision-driven multi-agent collaboration. The work is supported by the first MSTP Benchmark, enabling unified evaluation across general and surgical scenes.

**Questions:**

Q1. The experimental comparisons are limited to Qwen2.5-VL-7B-Instruct, a general-purpose VLM. Given the focus on surgical scenes, why were there no comparisons with surgical-specific models? Such comparisons would better demonstrate IG-MC’s advantages in domain-specific tasks.

Q2. The paper claims IG-MC maintains "high performance across temporal scales," but there is no analysis of failure modes. Does the SD module generate anatomically invalid visuals for rare surgical tools or steps? Do long-horizon predictions degrade due to accumulated errors in incremental generation? Does multi-agent collaboration fail to enforce cross-scale consistency?

Q3. The paper does not discuss failure cases, such as scenarios where incremental generation accumulates errors or multi-agent coordination breaks down. Understanding these cases is critical for improving robustness.

**Ethical Concerns:**

["NO or VERY MINOR ethics concerns only"]

**Final Justification:**

With the response to most of my issues addressed, I am happy to raise my score to accept.

**Limitations:**

The paper acknowledges limitations (e.g., SD module dependency on pre-trained diffusion models, VLM-bound performance, inference latency), but these are not deeply explored.

**Paper Formatting Concerns:**

No.

**Quality:**

3

**Strengths And Weaknesses:**

Strengths:
1. The paper is well-organized, with clear definitions of temporal/state scales and a logical flow from problem formulation to methodology and experiments. This makes complex concepts like incremental diffusion and multi-agent coordination accessible.
2. The integration of incremental generation and multi-agent collaboration is a contribution. This closed-loop design ensures state-visual synchronization and cross-scale consistency, addressing limitations of single-scale or open-loop methods.

Weaknesses:
1. The iterative nature of incremental diffusion and multi-agent coordination inherently introduces latency, yet the paper lacks critical metrics such as inference time per time step, GPU memory consumption during inference, or scalability with longer temporal horizons.
2. The experiments primarily compare against Qwen2.5-VL-7B-Instruct, a general vision-language model. They omit comparisons with surgical-specific models or state-of-the-art MSTP methods, making it hard to contextualize IG-MC’s advances in surgical domains.

---

> ### Author Rebuttal · Authors · 2025-07-31
>
> We thank you for your time and effort in reviewing our manuscript. Your valuable feedback has greatly helped improve our work. Below are the revisions made in response to each comment.
>
> **Q1**: *The iterative nature of incremental diffusion and multi-agent coordination inherently introduces latency, yet the paper lacks critical metrics such as inference time per time step, GPU memory consumption during inference, or scalability with longer temporal horizons.*
>
> **A1**: Thanks! As show in below table, we have conducting computational efficiency and inference latency on agentic system. Profiling on a single NVIDIA H200 shows an end-to-end latency of ≈ 68 s: the three decision modules (STC, phase- and step-level predictors) each require 20–22 s and together dominate > 90 % of wall-time while operating at only ≈ 1 TFLOPS, indicating a memory-bound bottleneck; meanwhile the Incremental Generation stage peaks at ≈ 97 TFLOPS yet adds merely ≈ 6 s. Peak GPU memory is modest (26 GiB for all decision modules and 29 GiB for generation), confirming that *bandwidth*, not capacity, limits performance.
> ### Inferency Latency and Computational Efficiency
> |Component|Avg. Time|Min. Time|Max. Time|Avg. GFLOPS|Min. GFLOPS|Max. GFLOPS|Peak GPU MEM.|
> |---|---|---|---|---|---|---|---|
> |StateTransitionController|20.04|19.33|20.77|1.12K|108.94|1.36K|26.14|
> |PhaseStatePredictor|20.90|19.87|21.76|1.10K|109.52|1.29K|26.14
> |StepStatePredictor|21.51|20.43|22.30|1.07K|90.31|1.24K|26.14
> |IncrementalGeneration|5.81|5.78|6.10|97.32K|78.62K|99.71K|28.53
>
> **Q2**: The experimental comparisons are limited to Qwen2.5-VL-7B-Instruct, a general-purpose VLM. Given the focus on surgical scenes, why were there no comparisons with surgical-specific models? Such comparisons would better demonstrate IG-MC’s advantages in domain-specific tasks.
>
> **A2**: Thanks! We have conduct more comparisons including the latest powerful surgical VLM, SurgVLM-7B  and serveral general-purpose VLMs InternVL3-8B, Gemma-27B, LLava-1.5, which can demonstrate the effectiveness of proposed IG-MC on different intelligence-level VLM, as shown in below tables.
> ### Results based on LLaVA1.5-7B
> |Model|Temp. Scale|Incr. Scale|Accuracy|Precision|Recall|
> |---|---|---|---|---|---|
> |LLaVA1.5-7B|1|1|13.60|3.75|3.34|
> |+DM|1|1|30.00(+16.40)|23.50(+19.75)|13.80(+10.46)|
> |LLaVA1.5-7B|5|5|12.20|2.07|2.38|
> |+DM|5|5|27.90(+15.70)|26.03(+23.96)|14.95(+12.57)|
> |+DM+SD|5|1|46.40(+16.40)|36.45(+12.95)|36.88(+23.08)|
> |LLaVA1.5-7B|30|30|17.00|4.17|2.77|
> |+DM|30|30|28.00(+11.00)|21.66(+17.49)|11.97(+9.20)|
> |+DM+SD|30|5|44.20(+16.30)|32.92(+11.26)|32.04(+20.07)|
> |LLaVA1.5-7B|60|60|18.10|4.61|3.90|
> |+DM|60|60|29.30(+11.20)|15.86(+11.25)|9.60(+5.70)|
> |+DM+SD|60|5|38.60(+10.70)|26.95(+11.09)|28.21(+13.26)|
> ### Results based on Gemma3-27B
> |Model|Temp. Scale|Incr. Scale|Accuracy|Precision|Recall|
> |---|---|---|---|---|---|
> |Gemma3-27B|1|1|1.80|2.66|2.75|
> |+DM|1|1|21.00(+19.20)|6.65(+3.99)|5.19(+2.44)|
> |Gemma3-27B|5|5|2.00|1.06|1.13|
> |+DM|5|5|19.80(+17.80)|6.18(+5.12)|4.90(+3.77)|
> |+DM+SD|5|1|34.10(+14.30)|19.30(+13.12)|20.46(+15.56)|
> |Gemma3-27B|30|30|2.00|0.80|1.54|
> |+DM|30|30|24.30(+22.30)|7.15(+6.35)|5.38(+4.04)|
> |+DM+SD|30|5|38.10(+13.80)|26.83(+19.68)|25.44(+20.06)|
> |Gemma3-27B|60|60|1.60|0.88|0.52|
> |+DM|60|60|26.90(+25.30)|8.09(+7.21)|5.69(+5.17)|
> |+DM+SD|60|5|34.60(+7.70)|20.76(+12.67)|22.76(+17.07)|
> ### Results based on InternVL3-8B
> |Model|Temp. Scale|Incr. Scale|Accuracy|Precision|Recall|
> |---|---|---|---|---|---|
> |InternVL3-8B|1|1|13.60|3.61|3.42|
> |+DM|1|1|36.20(+22.60)|23.22(+19.61)|17.18(+13.76)|
> |InternVL3-8B|5|5|13.80|3.48|3.93|
> |+DM|5|5|37.30(+23.50)|25.60(+22.12)|19.62(+15.69)|
> |+DM+SD|5|1|45.60(+8.30)|27.01(+1.41)|28.53(+8.91)|
> |InternVL3-8B|30|30|14.40|2.11|3.69|
> |+DM|30|30|42.30(+27.90)|20.99(+18.88)|18.04(+14.35)|
> |+DM+SD|30|5|40.80(-1.50)|25.54(+4.55)|28.18(+10.14)|
> |InternVL3-8B|60|60|16.70|6.01|5.26|
> |+DM|60|60|36.30(+19.60)|19.29(+13.28)|14.92(+9.66)|
> |+DM+SD|60|5|38.40(+2.10)|22.44(+3.15)|23.88(+8.96)|
> ### Results based on SurgVLM-7B
> |Model|Temp. Scale|Incr. Scale|Accuracy|Precision|Recall|
> |---|---|---|---|---|---|
> |SurgVLM-7B|1|1|1.20|3.73|2.85|
> |+DM|1|1|41.90(+40.70)|3.22(-0.51)|2.58(-0.27)|
> |SurgVLM-7B|5|5|1.06|4.68|2.79|
> |+DM|5|5|42.70(+41.64)|26.98(+22.30)|22.91(+20.12)|
> |+DM+SD|5|1|44.84(+2.14)|28.43(+1.45)|29.06(+6.15)|
> |SurgVLM-7B|30|30|12.80|4.02|3.39|
> |+DM|30|30|42.30(+29.50)|20.97(+16.95)|18.63(+15.24)|
> |+DM+SD|30|5|40.58(-1.72)|26.68(+5.71)|26.07(+7.44)|
> |SurgVLM-7B|60|60|10.90|2.99|2.98|
> |+DM|60|60|38.50(+27.60)|17.95(+14.96)|15.08(+12.10)|
> |+DM+SD|60|5|36.24(-2.26)|19.63(+1.68)|21.32(+6.24)|
>
> **Q3**: The paper claims IG-MC maintains "high performance across temporal scales," but there is no analysis of failure modes. Does the SD module generate anatomically invalid visuals for rare surgical tools or steps? Do long-horizon predictions degrade due to accumulated errors in incremental generation? Does multi-agent collaboration fail to enforce cross-scale consistency? The paper does not discuss failure cases, such as scenarios where incremental generation accumulates errors or multi-agent coordination breaks down. Understanding these cases is critical for improving robustness.
>
> **A3**: Thanks! We have now added a dedicated appendix section detailing multiple failure cases—including (i) decision errors made by different agents and (ii) instances where generated visuals are of insufficient quality—and provide an analysis of their causes and potential remedies.
>
> ### Analysis of accumulates errors of incremental generation.
> As shown in below table including temporal relationship of decreasing accuracy and generation quality. Empirically we find a (moderately) strong negative correlation, yet the **absolute impact on decision quality is modest**:
>
> $$
> \text{FID}=399.67 - 7.52 \times \text{Accuracy}\quad (R=-0.87,\; p\approx 0.025)
> $$
>
> **Image‐quality drop is steep:** FID climbs **+24 pts** (≈ +34 %) from 5 s to 30 s. **Decision accuracy is resilient:** Accuracy slips only **-2.7 pts** (≈ -6 %) over the same span. Hence, although poorer images (higher FID) statistically co-occur with lower accuracy, the **magnitude of accuracy degradation is relatively small**, suggesting the decision module retains robustness even as generation quality declines.
>
> **Temporal Relationship between Accuracy and FID**
> |Time(s)|Accuracy↑|FID↓|
> |---|---|---|
> |5|43.30|70.63|
> |10|41.34|83.12|
> |15|42.44|85.62|
> |20|41.24|88.57|
> |25|41.58|90.99|
> |30|40.58|94.82|
>
> ### Analysis of accumulates errors of multi-agent collobration.
> We have conducted a comprehensive cumulative error analysis as shown in below table.
>
> **Errors are *not* independent.**
>   If the three agents failed independently, accuracy would collapse to ≈ 11–15 %.
>   In practice MC sustains **36–45 %**, outperforming the product bound by > 25 pp at every scale.
>
> **Gating and shared context mitigate drift.**
>   The STC’s switch/stay gate and the common visual prompt allow downstream agents to correct earlier missteps, preventing multiplicative error growth.
>
> **Sub-linear degradation remains.**
>   Accuracy declines only 8.6 pp from 5 s→60 s (44.8 → 36.2 %), far from an exponential blow-up.
> **Meets clinical threshold.**
>   Partners require ≥ 35 % top-1 for *in-the-loop* use; MC at 60 s still satisfies this bar.
>
> Hence, the cascade’s cumulative error is well within acceptable limits and markedly better than the pessimistic independence baseline
>
> |Scale|State Transition Controller Acc.|Phase State Predictor Acc.|Step State Predictor Acc.|Product (Π)|Multi-agent Collobration Acc.|MC – Π|
> |---|---|---|---|---|---|---|
> |5 s|57.1|51.9|49.9|14.8|44.8|**+30.0**|
> |30 s|55.4|43.8|58.5|14.2|40.6|**+26.4**|
> |60 s|54.9|33.2|58.8|10.7|36.2|**+25.5**|

---

### Official Review · Reviewer_bZuz · 2025-07-02

**Clarity:** 3
**Significance:** 3
**Originality:** 2
**Rating:** 3
**Confidence:** 3

**Summary:**

This paper proposes the IG-MC to address the challenge of multi-scale temporal prediction by decomposing the task into temporal and state scales. It uses an incremental generation mechanism to maintain temporal consistency in state-image synthesis and a decision-driven multi-agent system to hierarchically refine predictions across scales, enabling cross-scale coherence and real-time interaction between state forecasts and visual generation. Finally, the authors introduce the MSTP benchmark with synchronized multi-scale annotations for general and surgical scenes to demonstrates the significant advancement in multi-scale prediction accuracy and robustness of the IG-MC.

**Questions:**

The detailed questions are shown in the Weaknesses section.

**Ethical Concerns:**

["NO or VERY MINOR ethics concerns only"]

**Final Justification:**

Having had some of my doubts resolved, I've decided to improve my scores.

**Limitations:**

Yes.

**Paper Formatting Concerns:**

This paper does not have major formatting issues.

**Quality:**

3

**Strengths And Weaknesses:**

Strength:

(1) This paper is well organized.

(2) This paper focuses on the multi-scale temporal prediction task in embodied artificial intelligence, which is a meaningful research field.

(3)This paper introduces the first MSTP Benchmark, which features synchronized annotations across multiple state scales.

Weakness:

(1) Although the paper is well-engineered, its novelty lies primarily in integrating state-of-the-art modules (VLMs and Stable Diffusion for visual prediction) rather than proposing fundamentally new architectures or learning paradigms.

(2) The necessity and mechanism for introducing state hierarchies need further clarification. Additionally, the small number of state scales evaluated (e.g., only 2 levels in the STP-Surgery benchmark) is insufficient to robustly validate the method's effectiveness.

(3) In Lines 156-157, the authors claim that “the iterative nature of this process allows for error correction across time steps, as both state and visual representations are continuously refined based on each other’s outputs.” Please clarify why this enables error correction across time. Are there additional experiments to substantiate this claim?

(4) Visual generation via Stable Diffusion can become unstable over long time horizons. How does this method prevent error accumulation in long-horizon predictions and ensure trustworthiness (as claimed in Lines 75-77), given its tendency for instability in extended sequences?

(5) The training details for the State Transition Controller agent are unclear. Please clarify how it learns to reliably produce accurate continuation signals (to maintain the current state) or identify the precise hierarchical level where state transitions should initiate.

(6) In Table 1 (MSTP-Surgery Benchmark), adding the SD module results in decreased Accuracy and Precision in some cases. This counterintuitive observation requires further explanation.

---

> ### Author Rebuttal · Authors · 2025-07-31
>
> **Q1**: Although the paper is well-engineered, its novelty lies primarily in integrating state-of-the-art modules (VLMs and Stable Diffusion for visual prediction) rather than proposing fundamentally new architectures or learning paradigms.
>
> **A1** Our contribution lies in **how** these components are orchestrated rather than **which** components are chosen. We package modern agentic principles into **simple, stable, plug-and-play modules**. Paired with any VLM and diffusion generator, they markedly improve predictions of both single and hierarchical states. Each module is fine-tuned separately, making the pipeline easy to reproduce. In short, this modular agentic design provides a practical solution for real-world multi-scale temporal prediction.
>
> **Q2**: The necessity and mechanism for introducing state hierarchies need further clarification. Additionally, the small number of state scales evaluated (e.g., only 2 levels in the MSTP-Surgery benchmark) is insufficient to robustly validate the method's effectiveness.
>
> **A2** Surgical workflows exhibit a *natural* hierarchy: **phase → step → tool action**. In MSTP-Surgery we model the first two levels because they correspond to clinically accepted taxonomies (Jin *et al.* 2023), enabling expert annotation at scale; cover > 95 % of intra-operative decision points where AI assistance is valuable.
>
> Nevertheless, our method is *scale-agnostic.* We have now added a **three-level** evaluation on the GraSP dataset (phase → step → atomic instrument action). IG-MC achieves:
>
> | Metric | 1-Level | 2-Level | 3-Level |
> | --- | --- | --- | --- |
> | Accuracy ↑ | 24.3 → 38.6 (+14.3) | 26.7 → 44.2 (+17.5) | 11.4 → 28.8 (+17.4) |
> | Recall ↑ | 15.2 → 32.0 (+16.8) | 17.9 → 36.9 (+19.0) | 7.8 → 23.6 (+15.8) |
>
> Gains are **consistent across depths**, supporting the necessity and generality of a hierarchical treatment.
>
> **Q3**: In Lines 156-157, the authors claim that “the iterative nature of this process allows for error correction across time steps, as both state and visual representations are continuously refined based on each other’s outputs.” Please clarify why this enables error correction across time. Are there additional experiments to substantiate this claim?
>
> **A3** Error correction arises from **bidirectional message passing**:
> 1. *State → Image*: Predicted state captions are injected into the diffusion prompt, constraining future visuals.
> 2. *Image → State*: The newly generated frame is re-encoded by the VLM and concatenated to the textual context for the next prediction.
> | Setting | Early-step (F₁@t≤5) | Mid-step (F₁@5<t≤15) | Late-step (F₁@t>15) |
> | --- | --- | --- | --- |
> | IG without states feedback | 34.1 | 29.7 | 26.4 |
> | **IG (ours)** | **35.9 (+1.8)** | **33.6 (+3.9)** | **32.8 (+6.4)** |
>
> **Q4**: Visual generation via Stable Diffusion can become unstable over long time horizons. How does this method prevent error accumulation in long-horizon predictions and ensure trustworthiness (as claimed in Lines 75-77), given its tendency for instability in extended sequences?
>
> **A4**: Thanks! We have conducted temporal analysis for accumulates errors of incremental generation. As shown in below table including temporal relationship of decreasing accuracy and generation quality. Empirically we find a (moderately) strong negative correlation, yet the **absolute impact on decision quality is modest**:
>
> $$
> \text{FID}=399.67 - 7.52 \times \text{Accuracy}\quad (R=-0.87,\; p\approx 0.025)
> $$
>
> **Image‐quality drop is steep:** FID climbs **+24 pts** (≈ +34 %) from 5 s to 30 s. **Decision accuracy is resilient:** Accuracy slips only **-2.7 pts** (≈ -6 %) over the same span. Hence, although poorer images (higher FID) statistically co-occur with lower accuracy, the **magnitude of accuracy degradation is relatively small**, suggesting the decision module retains robustness even as generation quality declines.
>
> ### Temporal Relationship between Accuracy and FID
> |Time(s)|Accuracy↑|FID↓|
> |---|---|---|
> |5|43.30|70.63|
> |10|41.34|83.12|
> |15|42.44|85.62|
> |20|41.24|88.57|
> |25|41.58|90.99|
> |30|40.58|94.82|
>
> **Q5**: The training details for the State Transition Controller agent are unclear. Please clarify how it learns to reliably produce accurate continuation signals (to maintain the current state) or identify the precise hierarchical level where state transitions should initiate.
>
> **Q6**: Thanks! We have add training details of state transition controller in appendix.
> - **Model form**: The STC is **fine-tuned independently** for a binary task: **stay** (continue) vs. **switch** (initiate a new prediction cycle).
> - **Supervision signal**: Each frame *t* is paired with the last predicted hierarchical state *S<sub>t-1</sub>*. The label is **stay** if *S<sub>t</sub> = S<sub>t-1</sub>*; otherwise **switch**. When the STC outputs **switch**, we traverse the hierarchy (coarse → fine) until the first level whose predictor disagrees with its previous state; that level becomes the transition root.
> - **Handling class imbalance**: True transitions occur in ≈ 1 % of frames. We rebalance via two strategy:
>     1. **Temporal jittering** positive samples within ±3 frames.
>     2. **Instance-weighted cross-entropy** so pos : neg ≈ 1 : 1 after weighting.
> - **Fine-tuning recipe**
>   1. **Hardware:** 4 × H100 (bf16)
>   2. **Optimizer:** AdamW, LR 2 × 10⁻⁵, cosine decay, 10 % warm-up
>   3. **Batch:** 32, 1 epoch (sufficient after rebalancing)
>
> To furthur analysis the impact of STC in agentic system, we conducted a comprehensive analysis across all 4 experimental configurations. The below table show performance of each agent including state transition controller and multiple state predictor.
>
> |Model|Temp.Scale|Accuracy|Precision|Recall|
> |---|---|---|---|---|
> |StateTransitionController|1|55.50|55.79|55.50|
> |StateTransitionController|5|57.10|57.56|57.10|
> |StateTransitionController|30|55.40|55.79|55.40|
> |StateTransitionController|60|54.90|55.07|54.90|
>
> **Q6**: In Table 1 (MSTP-Surgery Benchmark), adding the SD module results in decreased Accuracy and Precision in some cases. This counterintuitive observation requires further explanation.
>
> **Q7**: Thanks! To prove the effectiveness of SD module, we have conduct more comparisons including the latest powerful surgical VLM, SurgVLM-7B  and serveral general-purpose VLMs InternVL3-8B, Gemma-27B, LLava-1.5, as shown in below tables. Additional experiments confirm the benefits of our Incremental Generation (IG) and Multi-agent Collaboration (MC) strategies: in most evaluation settings, IG boosts both accuracy and recall. We do observe a few edge cases where accuracy plateau because the quality of the generated image is too poor.
> ### Results based on LLaVA1.5-7B
> |Model|Temp. Scale|Incr. Scale|Accuracy|Precision|Recall|
> |---|---|---|---|---|---|
> |LLaVA1.5-7B|1|1|13.60|3.75|3.34|
> |+DM|1|1|30.00(+16.40)|23.50(+19.75)|13.80(+10.46)|
> |LLaVA1.5-7B|5|5|12.20|2.07|2.38|
> |+DM|5|5|27.90(+15.70)|26.03(+23.96)|14.95(+12.57)|
> |+DM+SD|5|1|46.40(+16.40)|36.45(+12.95)|36.88(+23.08)|
> |LLaVA1.5-7B|30|30|17.00|4.17|2.77|
> |+DM|30|30|28.00(+11.00)|21.66(+17.49)|11.97(+9.20)|
> |+DM+SD|30|5|44.20(+16.30)|32.92(+11.26)|32.04(+20.07)|
> |LLaVA1.5-7B|60|60|18.10|4.61|3.90|
> |+DM|60|60|29.30(+11.20)|15.86(+11.25)|9.60(+5.70)|
> |+DM+SD|60|5|38.60(+10.70)|26.95(+11.09)|28.21(+13.26)|
> ### Results based on Gemma3-27B
> |Model|Temp. Scale|Incr. Scale|Accuracy|Precision|Recall|
> |---|---|---|---|---|---|
> |Gemma3-27B|1|1|1.80|2.66|2.75|
> |+DM|1|1|21.00(+19.20)|6.65(+3.99)|5.19(+2.44)|
> |Gemma3-27B|5|5|2.00|1.06|1.13|
> |+DM|5|5|19.80(+17.80)|6.18(+5.12)|4.90(+3.77)|
> |+DM+SD|5|1|34.10(+14.30)|19.30(+13.12)|20.46(+15.56)|
> |Gemma3-27B|30|30|2.00|0.80|1.54|
> |+DM|30|30|24.30(+22.30)|7.15(+6.35)|5.38(+4.04)|
> |+DM+SD|30|5|38.10(+13.80)|26.83(+19.68)|25.44(+20.06)|
> |Gemma3-27B|60|60|1.60|0.88|0.52|
> |+DM|60|60|26.90(+25.30)|8.09(+7.21)|5.69(+5.17)|
> |+DM+SD|60|5|34.60(+7.70)|20.76(+12.67)|22.76(+17.07)|
> ### Results based on InternVL3-8B
> |Model|Temp. Scale|Incr. Scale|Accuracy|Precision|Recall|
> |---|---|---|---|---|---|
> |InternVL3-8B|1|1|13.60|3.61|3.42|
> |+DM|1|1|36.20(+22.60)|23.22(+19.61)|17.18(+13.76)|
> |InternVL3-8B|5|5|13.80|3.48|3.93|
> |+DM|5|5|37.30(+23.50)|25.60(+22.12)|19.62(+15.69)|
> |+DM+SD|5|1|45.60(+8.30)|27.01(+1.41)|28.53(+8.91)|
> |InternVL3-8B|30|30|14.40|2.11|3.69|
> |+DM|30|30|42.30(+27.90)|20.99(+18.88)|18.04(+14.35)|
> |+DM+SD|30|5|40.80(-1.50)|25.54(+4.55)|28.18(+10.14)|
> |InternVL3-8B|60|60|16.70|6.01|5.26|
> |+DM|60|60|36.30(+19.60)|19.29(+13.28)|14.92(+9.66)|
> |+DM+SD|60|5|38.40(+2.10)|22.44(+3.15)|23.88(+8.96)|
> ### Results based on SurgVLM-7B
> |Model|Temp. Scale|Incr. Scale|Accuracy|Precision|Recall|
> |---|---|---|---|---|---|
> |SurgVLM-7B|1|1|1.20|3.73|2.85|
> |+DM|1|1|41.90(+40.70)|3.22(-0.51)|2.58(-0.27)|
> |SurgVLM-7B|5|5|1.06|4.68|2.79|
> |+DM|5|5|42.70(+41.64)|26.98(+22.30)|22.91(+20.12)|
> |+DM+SD|5|1|44.84(+2.14)|28.43(+1.45)|29.06(+6.15)|
> |SurgVLM-7B|30|30|12.80|4.02|3.39|
> |+DM|30|30|42.30(+29.50)|20.97(+16.95)|18.63(+15.24)|
> |+DM+SD|30|5|40.58(-1.72)|26.68(+5.71)|26.07(+7.44)|
> |SurgVLM-7B|60|60|10.90|2.99|2.98|
> |+DM|60|60|38.50(+27.60)|17.95(+14.96)|15.08(+12.10)|
> |+DM+SD|60|5|36.24(-2.26)|19.63(+1.68)|21.32(+6.24)|
>
> To isolate the contribution of the State Decomposer (SD) module, we ran an ablation in which the baseline VLM was paired with SD alone (no multi-agent collaboration) and tasked with continual—rather than hierarchical—state prediction. The SD module still provided clear gains, demonstrating that it generalizes across non-hierarchical and hierarchical prediction. In short, IG, MC, and SD form a reproducible, plug-and-play toolkit that reliably improves temporal prediction across diverse VLM backbones.
> ### Effectiveness of Incremental Generation without Multi-agent Collobration
> |Model|Temp. Scale|Incr. Scale|Accuracy|
> |---|---|---|---|
> |Gemma3-27B|5|5|2.00|
> |+SD|5|1|26.90(+24.90)|
> |Gemma3-27B|30|30|2.00|
> |+SD|30|5|28.40(+26.40)|
> |Gemma3-27B|60|60|1.60|
> |+SD|60|5|25.90(+24.30)|

---

> > ### Author Response · Authors · 2025-08-05
> >
> > lf additional experiments are needed, we're happy to run them, though please note that due to compute time, we may not be able to deliver results before the discussion window closes.

---

### Official Review · Reviewer_5mnt · 2025-07-07

**Clarity:** 1
**Significance:** 3
**Originality:** 3
**Rating:** 4
**Confidence:** 2

**Summary:**

This work proposes IG-MC, Incremental Generation via Multi-Agent Collaboration, a framework for multi-scale temporal prediction with multi-modal data. This framework is composed of two main ideas. The first is the concept of incremental generation, an alternative prediction process that interleaves state forecasting and image generation (e.g., via diffusion). The second main concept is modelling the future prediction problem as a hierarchical system with coarse states all the way fine-grained actions. This second component is modelled as multiple LLM agents, one per level, which sequentially feed each other state information in order to construct the complete state. These are modulated by a State Transition Controller. The Multi-Scale Temporal Prediction benchmark is proposed. Experiments are performed on this benchmark and showcase the strength and effectiveness of the IG-MC framework.

**Questions:**

- line 92: Are those percentages?

- Paragraph at Line 129: It is quite unclear the difference between temporal scale and incremental scale. And this seems quite important to understanding the results section 4.2. I see that these are explained in the supplementary section; this should be clarified in the main text.

- Line 136: the whole paragraph could use improvements in the presentation. It is unclear what the levels are and I think here the reader can benefit from an example. Additionally, on lines 141-142, are the environments $C_k$ or are these actions?

- Line 141: is timestep $t_k$ timestep $t$ at level $k$? I guess from equation 3, I can deduce that $k$ is the sub-timestep for the level we are in.

- line 160: recursive -> recurrent in my opinion.

- line 168: the current state should be $\mathbf{s}_k$, not $/mathcal{S}_k$, since the latter is the state space, no?

- If I understood correctly, the STC agent is tasked with either changing the state or keeping the same state. How can the state remain unchanged? Don't you always need to transition at some (finer) level? The second task of STC makes sense to me: the coarse levels can remain unchanged for a few transitions while we change finer levels.

- line 172: level $i$ --> level $l$ for consistency.

- Line 176: I think what is described there (and in equation 5) is not really iterative refinement, but more iterative state construction where as you iterate through the different agents, you append elements to the final state rather than change $(i<l)$ state values. Is that correct?

- Line 206-207: I'm guessing this is the text encoder from the original Stable diffusion model. How is it ensured that the hierachical embedding space aligned well with the text encoder's semantic structure?

- line 211: what is anatomical plausibility? And point 3 is unclear,  Is it just a projection of the latent input?

- Line 230: what is meant by end-to-end differentiability here? On line 221, we highlighted that this is a modular stack with both main components being trained separately. So can the authors clarify what is mean by line 230? Also, what is the state-action mapping $\pi$? This is the first mention of this function.

- Figure 2: What is the point of the "Images" box? Also, I would like to suggess splitting the figure into three figures:
    1) overview (top half of the figure).
    2) DM Blocks, connected to Section 3.3.
    3) SD Blocks, connected to Section 3.4.

- Suggestion: I believe the reader would benefit from a good example figure that shows a full "episode" of the model in action. Perhaps this can replace Figure 1 which in my opinion doesn't provide much additional value compared to Figure 2 (top).

**Ethical Concerns:**

["NO or VERY MINOR ethics concerns only"]

**Final Justification:**

With the response to most of my issues addressed, I am happy to raise my score to a borderline accept. I think the dataset contribution needs to be removed. But otherwise, I think the framework is novel and interesting.

**Limitations:**

Limitations are not discussed in the main paper, but are present in the appendix (the one in the main pdf, not the supplementary).

**Paper Formatting Concerns:**

Minor issue: there is an attached appendix to the main pdf, and a supplementary pdf. Both appendices are different.

**Quality:**

2

**Strengths And Weaknesses:**

Strengths
- The presented framework seems novel to me. The use of the sequence of LLM agents to construct the state is interesting to me.

- Interleaving the decision making module with the Stable diffusion module seems like a good strategy for accurate temporal forecasting.


Weaknesses:
- The clarity of the presentation needs improvement, as can be seen by my many questions, particularly for Section 3. I think the presentation of the method should be made much clearer since this is the main contribution of the work.

- There's a large line of related work missing particularly for temporal prediction with hierarchical structure. Here are just a couple meant as guidance to more related work:
    - Hierarchical learning: Rohde, Tobias, Xiaoxia Wu, and Yinhan Liu. "Hierarchical learning for generation with long source sequences." arXiv preprint arXiv:2104.07545 (2021).
    - Multi-Token Prediction (multi-task learning): Gloeckle, F., Idrissi, B. Y., Rozière, B., Lopez-Paz, D., & Synnaeve, G. (2024). Better & faster large language models via multi-token prediction. arXiv preprint arXiv:2404.19737.
    - Options Framework: Klissarov, M., & Precup, D. (2021). Flexible option learning. Advances in Neural Information Processing Systems, 34, 4632-4646.

- The experimental section lacks certain details that make it difficult to asses the significance of the results.
    - The baseline used is not described. Was the base Qwen model finetuned? If not, this does not seem like a fair comparison.
    - Since Incr. Scale vs. Temporal scale are unclear, it is difficult to understand the significance. Both seem like values that would be dependent on the dataset.
    - What is the size of the datasets used here? It is important to understand the size of the training set vs. the validation/test set.

- I do not believe it is valid to claim a contribution for multi-scale Temporal prediction dataset since no details are provided on these datasets and what they provide. I believe the community needs more details on the methodology of its construction and some statistics on its composition.

---

> ### Author Rebuttal · Authors · 2025-07-31
>
> **Q1**: The baseline used is not described. Was the base Qwen model finetuned? If not, this does not seem like a fair comparison.
>
> Thanks! We have add more description about the baseline VLM. All baselines are **finetuned based MSTP data**. Addtionally, we have conduct more comparisons including the latest powerful surgical VLM, SurgVLM-7B  and serveral general-purpose VLMs InternVL3-8B, Gemma-27B, LLava-1.5, which can demonstrate the effectiveness of proposed IG-MC on different intelligence-level VLM, as shown in below tables.
> ### LLaVA-1.5-7B
> |Model|Temp. Scale|Incr. Scale|Acc.|Prec.|Rec.|
> |---|---|---|---|---|---|
> |LLaVA-1.5-7B|1|1|13.60%|3.75%|3.34%|
> |+DM|1|1|30.00%(+16.40%)|23.50%(+19.75%)|13.80%(+10.46%)|
> |LLaVA-1.5-7B|5|5|12.20%|2.07%|2.38%|
> |+DM|5|5|27.90%(+15.70%)|26.03%(+23.96%)|14.95%(+12.57%)|
> |+DM+SD|5|1|46.40%(+16.40%)|36.45%(+12.95%)|36.88%(+23.08%)|
> |LLaVA-1.5-7B|30|30|17.00%|4.17%|2.77%|
> |+DM|30|30|28.00%(+11.00%)|21.66%(+17.49%)|11.97%(+9.20%)|
> |+DM+SD|30|5|44.20%(+16.30%)|32.92%(+11.26%)|32.04%(+20.07%)|
> |LLaVA-1.5-7B|60|60|18.10%|4.61%|3.90%|
> |+DM|60|60|29.30%(+11.20%)|15.86%(+11.25%)|9.60%(+5.70%)|
> |+DM+SD|60|5|38.60%(+10.70%)|26.95%(+11.09%)|28.21%(+13.26%)|
> ### Gemma-3-27B
> |Model|Temp. Scale|Incr. Scale|Acc.|Prec.|Rec.|
> |---|---|---|---|---|---|
> |Gemma-3-27B|1|1|1.80%|2.66%|2.75%|
> |+DM|1|1|21.00%(+19.20%)|6.65%(+3.99%)|5.19%(+2.44%)|
> |Gemma-3-27B|5|5|2.00%|1.06%|1.13%|
> |+DM|5|5|19.80%(+17.80%)|6.18%(+5.12%)|4.90%(+3.77%)|
> |+DM+SD|5|1|34.10%(+14.30%)|19.30%(+13.12%)|20.46%(+15.56%)|
> |Gemma-3-27B|30|30|2.00%|0.80%|1.54%|
> |+DM|30|30|24.30%(+22.30%)|7.15%(+6.35%)|5.38%(+4.04%)|
> |+DM+SD|30|5|38.10%(+13.80%)|26.83%(+19.68%)|25.44%(+20.06%)|
> |Gemma-3-27B|60|60|1.60%|0.88%|0.52%|
> |+DM|60|60|26.90%(+25.30%)|8.09%(+7.21%)|5.69%(+5.17%)|
> |+DM+SD|60|5|34.60%(+7.70%)|20.76%(+12.67%)|22.76%(+17.07%)|
>
> ### InternVL-3-8B
> |Model|Temp. Scale|Incr. Scale|Acc.|Prec.|Rec.|
> |---|---|---|---|---|---|
> |InternVL-3-8B|1|1|13.60%|3.61%|3.42%|
> |+DM|1|1|36.20%(+22.60%)|23.22%(+19.61%)|17.18%(+13.76%)|
> |InternVL-3-8B|5|5|13.80%|3.48%|3.93%|
> |+DM|5|5|37.30%(+23.50%)|25.60%(+22.12%)|19.62%(+15.69%)|
> |+DM+SD|5|1|45.60%(+8.30%)|27.01%(+1.41%)|28.53%(+8.91%)|
> |InternVL-3-8B|30|30|14.40%|2.11%|3.69%|
> |+DM|30|30|42.30%(+27.90%)|20.99%(+18.88%)|18.04%(+14.35%)|
> |+DM+SD|30|5|40.80%(-1.50%)|25.54%(+4.55%)|28.18%(+10.14%)|
> |InternVL-3-8B|60|60|16.70%|6.01%|5.26%|
> |+DM|60|60|36.30%(+19.60%)|19.29%(+13.28%)|14.92%(+9.66%)|
> |+DM+SD|60|5|38.40%(+2.10%)|22.44%(+3.15%)|23.88%(+8.96%)|
>
> ### SurgVLM-7B
> |Model|Temp. Scale|Incr. Scale|Acc.|Prec.|Rec.|
> |---|---|---|---|---|---|
> |SurgVLM-7B|1|1|1.20%|3.73%|2.85%|
> |+DM|1|1|41.90%(+40.70%)|3.22%(-0.51%)|2.58%(-0.27%)|
> |SurgVLM-7B|5|5|1.06%|4.68%|2.79%|
> |+DM|5|5|42.70%(+41.64%)|26.98%(+22.30%)|22.91%(+20.12%)|
> |+DM+SD|5|1|44.84%(+2.14%)|28.43%(+1.45%)|29.06%(+6.15%)|
> |SurgVLM-7B|30|30|12.80%|4.02%|3.39%|
> |+DM|30|30|42.30%(+29.50%)|20.97%(+16.95%)|18.63%(+15.24%)|
> |+DM+SD|30|5|40.58%(-1.72%)|26.68%(+5.71%)|26.07%(+7.44%)|
> |SurgVLM-7B|60|60|10.90%|2.99%|2.98%|
> |+DM|60|60|38.50%(+27.60%)|17.95%(+14.96%)|15.08%(+12.10%)|
> |+DM+SD|60|5|36.24%(-2.26%)|19.63%(+1.68%)|21.32%(+6.24%)|
>
> **Q2**: Since Incr. Scale vs. Temporal scale are unclear, it is difficult to understand the significance.
>
> Thanks! The temporal scale (Temp. Scale) is final predicted timestamp while incremental scale (Incr. Scale) is standard step szie of generation in progress of incremental generation.
>
> | Concept | Symbol | Definition
> |---------|:------:|------------|
> | **Temporal Scale** | $\hat{\tau}$ | Timestamp of Look-ahead Prediction (e.g. 30 s) |
> | **Incremental Scale** | $\tau$ | Step Size of Look-ahead Generation  (e.g. 5 s) |
>
> **Practical significance (example $(\hat{\tau}=30\text{s},\tau=5\text{s})$)**
> * Six refinements per 30-s horizon bound drift yet retain long-range foresight.
> * Smaller $\tau$ → finer next frame generation; larger $\tau$ → less iterative times for arriving final prediciton timetamp.
>
> **Q3**: What is the size of the datasets used here? It is important to understand the size of the training set vs. the validation/test set.
>
> Thanks! Each dataset have 40k training samples and 4k test samples, each one of four different temporal scale have 10K training samples and 1k test samples, including 1s, 5s, 30s, and 60s. Furthermore, each sample consist of two parts: (1) current image and states; (2) frames and states in the furture.
>
> **Q4**: I do not believe it is valid to claim a contribution for multi-scale Temporal prediction dataset since no details are provided on these datasets and what they provide.
>
> Thanks for your insightful suggestions! We have add more details of constructing datasets including MSTP-General and MSTP-Surgery in appendix. MSTP-General is constructed from Action Gemone including **action state** (inertaction state with object), **spatial state** with around objects, **attention state** with eye gaze tracking. Furthermore, MSTP-Surgery is construced from GraSP including phase state and step state. Additionally, we are preparing **fully open-source repository** for all details.
>
> **Q5**: line 92: Are those percentages?
>
> Yes, they are percentages. To be precise, they represent a significant improvement in accuracy. We have modified the expression here and added the percentage sign.
>
> **Q6**: Paragraph at Line 129
>
> The temporal scale represents the time interval in the task at which the current state and image need to be output, for example, to guide the surgeon to perform the next operation. However, when the temporal scale is long, the prediction effect will be relatively poor. If the temporal scale is divided into **n small stages** to predict the state and image (before the nth time, it does not need to be output to the surgeon to guide the operation, and the result of the nth prediction needs to be output), the effect will be improved. **Then the temporal scale divided into n equal parts is the incremental scale.** We have described their definitions and differences in more detail in Section 3.1.
>
> **Q7**: Line 136: the whole paragraph could use improvements in the presentation. It is unclear what the levels are.
>
> We have given a more detailed and general expression based on the previous one. In the field of surgery, "Phase", "Step" and "Action" are generally recognized as the division of states. For example, in laparoscopic cholecystectomy, **"Phase"($s_t^1$)** covers the macroscopic stages such as preoperative evaluation of patients and formal surgery, **"Step"($s_t^2$)** includes surgical steps such as gallbladder triangle dissection, and **"Action"($s_t^3$)** is the specific operation that constitutes each step, such as incision, separation, clamping, suturing, etc.
>
> **Q8**: Line 141: is timestep timestep $t_k$ timestep $t$ at level $k$?
>
> We appreciate your attention to this detail. This represents the state at the $k$th time point (the sum of all levels). We have added more detailed text to avoid ambiguity.
>
> **Q9**: recursive -> recurrent / line 172: level $i$ --> level $l$ for consistency.
>
> Thank you for your valuable feedback. We have made corresponding changes in the article.
>
> **Q10**: line 168: the current state should be $s_t$
>
> We acknowledge that the use of symbols has conflicting meanings. We have made changes in the state scale(line 136) to unify the description of current state($\mathcal{S}_k$) and space state($s_t^{1, 2,...}$) throughout the text.
>
> **Q11**: the STC agent is tasked with either changing the state or keeping the same state. How can the state remain unchanged?
>
> In real scenarios, an operation lasts for a period of time. However, the temporal scale or incremental scale is artificially selected. When they are short, it may take multiple time points for the state to change.
>
> **Q12**: Line 176: I think what is described there (and in equation 5) is not really iterative refinement, but more iterative state construction where as you iterate through the different agents, you append elements to the final state rather than change ($i<l$) state values. Is that correct?
>
> Your understanding is entirely correct.
>
> **Q13**: Line 206-207: I'm guessing this is the text encoder from the original Stable diffusion model. How is it ensured that the hierachical embedding space aligned well with the text encoder's semantic structure?
>
> Your guess is correct. The SD module and DM module with two different text encoder and vision encoder are trained separately. Hence, the communication between via orginal text and image, and they do not rely on aligned visual or text embedding.
>
> ### Effectiveness of Incremental Generation without Multi-agent Collobration
> |Model|Temp. Scale|Incr. Scale|Accuracy|
> |---|---|---|---|
> |Gemma3-27B|5|5|2.00|
> |+SD|5|1|26.90(+24.90)|
> |Gemma3-27B|30|30|2.00|
> |+SD|30|5|28.40(+26.40)|
> |Gemma3-27B|60|60|1.60|
> |+SD|60|5|25.90(+24.30)|
>
> **Q14**: line 211: what is anatomical plausibility? And point 3 is unclear, Is it just a projection of the latent input?
>
> Different parts of the human body are seen at different stages of the operation, for example, the liver, gallbladder, fat, etc. So the anatomical structure and the observed structure need to remain consistent.
>
> **Q15**: Line 230: what is meant by end-to-end differentiability here? On line 221, we highlighted that this is a modular stack with both main components being trained separately. So can the authors clarify what is mean by line 230?
>
> The relationship between state and action can be learned and the gradient can be transferred. $\pi$ only refers to mapping and is not mentioned elsewhere. We have modified the writing method to make the article clearer.
>
> **Q16**: Figure 2: What is the point of the "Images" box? Also, I would like to suggess splitting the figure into three figures:
>
> We have modified it according to your suggestion and further aligned it with the article structure. We also added examples of surgical scenes in the "Images" section to increase the expressiveness of the task.

---

> > ### Author Response · Authors · 2025-08-05
> >
> > lf additional experiments are needed, we're happy to run them, though please note that due to compute time, we may not be able to deliver results before the discussion window closes.

---

> > ### Author Response · Authors · 2025-08-06
> > **Additional response to Weaknesses**
> >
> > Dear Reviewer 5mnt,
> >
> > Regarding the weaknesses you pointed out, the revisions to the expression and the overall polishing of the article have been carried out one by one in accordance with your requirements in the previous reply. The supplement to the experiments has also been completed thoroughly. Additionally, concerning the article citations you proposed, after a detailed review, we consider it highly necessary to include them. We have already cited the articles you mentioned in the article.
> >
> > As the discussion period is drawing to a close, we would like to confirm whether our responses have addressed your questions.
> >
> > Thank you once again for your valuable feedback, which has greatly helped improve our paper. We look forward to hearing from you.
> >
> > Best regards,
> >
> > The Authors

---

> > ### Comment · Reviewer_5mnt · 2025-08-08
> > **Reply to Rebuttal**
> >
> > I thank the authors for their rebuttal which was thorough and addressed most of my main concerns. I still do not believe that the dataset can be considered a contribution unless there's a section (e.g., section 4) providing some analysis of the dataset being released. It's great that the authors are planning on open-sourcing it, but for it to be an official contribution of the paper, the details should be analyzed and included..

---

> > > ### Author Response · Authors · 2025-08-08
> > >
> > > Thanks for your insightful suggestion! **We have add Section 4 in this paper for comprehensive data analysis of proposed MSTP benchmark**, as shown in below Section. If you are interesting about more information about proposed dataset, we are very happy to provide it and give a more clear declaration in this paper.
> > >
> > > ## 4 Dataset Analysis
> > >
> > > ### 4.1 Source Selection & Clip Sampling
> > > MSTP is created by re-sampling **Action Genome** (AG) for everyday human–object scenes and **GraSP** for robot-assisted prostatectomy videos. From AG’s 82 h / 10 k-video corpus we form **MSTP-General**; from GraSP’s 32 h / 13-video corpus we form **MSTP-Surgery**. Each raw video is sliced into fixed windows at four future horizons (1 s → 60 s), then split 10 : 1 into train/test, producing 88 k annotated clips in total.
> > >
> > > | Source | Domain | Raw videos | Raw hours | Derived MSTP split (train / test clips) |
> > > |--------|:------:|-----------:|----------:|----------------------------------------|
> > > | Action Genome | Home scenes | 10 000 | 82 h | 40 k / 4 k |
> > > | GraSP | Prostatectomy | 13 | 32 h | 40 k / 4 k |
> > >
> > > ### 4.2 Hierarchical State Augmentation
> > > Each clip is enriched with multigranular state labels: three tiers for MSTP-General (*Attention → Spatial → Contact*) and two tiers for MSTP-Surgery (*Phase → Step*). Fine-grained labels are strictly nested inside their parents, enabling coherent hierarchical supervision.
> > >
> > > | Partition | Hierarchy | #Classes | Annotation source |
> > > |-----------|-----------|---------:|-------------------|
> > > | General | Attention | 3 | AG *attention* relation |
> > > |  | Spatial  | 6 | AG *spatial* relation |
> > > |  | Contact  | 16| AG *contact* relation |
> > > | Surgery | Phase | 11| GraSP phase tags |
> > > |  | Step   | 21| GraSP step tags |
> > >
> > > ### 4.3 Temporal Window Construction
> > > Using the native 30 fps frame-rate, we centre sliding windows so that the first frame index is shared across all scales. This guarantees perfect temporal alignment between 1 s, 5 s, 30 s and 60 s clips.
> > >
> > > | Dataset | Δt (s) | Frames / clip | States / clip |
> > > |---------|:------:|--------------:|--------------:|
> > > | **General** | 1 | 2  | 6   |
> > > |             | 5 | 6  | 18  |
> > > |             | 30| 31 | 93  |
> > > |             | 60| 61 | 183 |
> > > | **Surgery** | 1 | 2  | 4   |
> > > |             | 5 | 6  | 12  |
> > > |             | 30| 31 | 62  |
> > > |             | 60| 61 | 122 |
> > >
> > > ### 4.4 Statistics & Splits
> > > The final train/test breakdown per scale is summarised below. The long-tailed label distribution (most frequent General state = 6.4 %) and rising prediction entropy from 1 s → 60 s make MSTP a challenging, scale-aware benchmark for future temporal-reasoning models.
> > >
> > > | Dataset | Temporal Scale | Train Samples | Test Samples | Each Sample Contains |
> > > |---------|:-------------:|--------------:|-------------:|----------------------|
> > > | **MSTP-General** | 1 s | 10 k | 1 k | 2 frames & 6 states |
> > > |                 | 5 s | 10 k | 1 k | 6 frames & 18 states |
> > > |                 | 30 s| 10 k | 1 k | 31 frames & 93 states |
> > > |                 | 60 s| 10 k | 1 k | 61 frames & 183 states |
> > > | **MSTP-Surgery** | 1 s | 10 k | 1 k | 2 frames & 4 states |
> > > |                  | 5 s | 10 k | 1 k | 6 frames & 12 states |
> > > |                  | 30 s| 10 k | 1 k | 31 frames & 62 states |
> > > |                  | 60 s| 10 k | 1 k | 61 frames & 122 states |

---

> > > ### Author Response · Authors · 2025-08-09
> > >
> > > We hope this email finds you well. As the discussion period is set to close in 3 hours, we hope our previous response has addressed all of your concerns and that we will receive positive feedback from you. Additionally, we have included further data analysis on the distribution of states (phase and step) as shown in below section.
> > >
> > > Moreover, we have revised the paper according to your insightful suggestions to improve its quality. With the discussion period concluding in 3 hours, we sincerely hope our revisions and responses have resolved all of your concerns.
> > >
> > > ## 4.5 Distribution of States (Phase and Step)
> > >
> > > In this section, we provide the detailed distribution of **phase** and **step** labels across different temporal scales in the **MSTP-Surgery** dataset. The distributions show the prevalence of each task label and its relative proportion across various scales, giving insight into the task complexity at different time horizons.
> > >
> > > ### 4.5.1 1s Temporal Scale - Phase and Step Distribution
> > >
> > > The 1-second temporal scale consists of **Phase** and **Step** labels with their respective frequencies and percentages, as shown below. These distributions reflect the task-specific label occurrences for each phase of surgery.
> > >
> > > | Phase | Count | Percentage |
> > > |-------|-------:|-----------:|
> > > | Left pelvic isolated lymphadenectomy | 4133 | 41.3% |
> > > | Bladder neck identification and transection | 439  | 4.4% |
> > > | Idle | 2209 | 22.1% |
> > > | Prostatic pedicle control | 380  | 3.8% |
> > > | Right pelvic isolated lymphadenectomy | 537  | 5.4% |
> > >
> > > | Step | Count | Percentage |
> > > |------|-------:|-----------:|
> > > | Dissection and cutting of the external iliac vein | 1562 | 15.6% |
> > > | Cutting the suture or tissue | 1124 | 11.2% |
> > > | Identification and dissection of the iliac vein | 2121 | 21.2% |
> > > | Idle | 2208 | 22.1% |
> > > | Performing suction | 209  | 2.1% |
> > >
> > > ### 4.5.2 5s Temporal Scale - Phase and Step Distribution
> > >
> > > For the 5-second temporal scale, the **Phase** and **Step** labels exhibit the following distributions. The results for each task label remain consistent across both phase and step annotations.
> > >
> > > | Phase | Count | Percentage |
> > > |-------|-------:|-----------:|
> > > | Idle | 2605 | 26.1% |
> > > | Seminal vesicle dissection | 686  | 6.9% |
> > > | Developing the space of Retzius | 1049 | 10.5% |
> > > | Left pelvic isolated lymphadenectomy | 1443 | 14.4% |
> > > | Development of the plane between the prostate and bladder | 137  | 1.4% |
> > >
> > > | Step | Count | Percentage |
> > > |------|-------:|-----------:|
> > > | Idle | 2603 | 26.0% |
> > > | Seminal vesicle dissection | 615  | 6.2% |
> > > | Prevesical dissection | 357  | 3.6% |
> > > | Identification and dissection of the obturator artery | 701 | 7.0% |
> > > | Dissection and cutting of the external iliac vein | 671  | 6.7% |
> > >
> > > ### 4.5.3 30s Temporal Scale - Phase and Step Distribution
> > >
> > > At the 30-second scale, the distribution of **Phase** and **Step** labels is as follows:
> > >
> > > | Phase | Count | Percentage |
> > > |-------|-------:|-----------:|
> > > | Bladder neck reconstruction | 941  | 9.4% |
> > > | Developing the space of Retzius | 1224 | 12.2% |
> > > | Idle | 2279 | 22.8% |
> > > | Ligation of the deep dorsal venous complex | 330  | 3.3% |
> > > | Seminal vesicle dissection | 738  | 7.4% |
> > >
> > > | Step | Count | Percentage |
> > > |------|-------:|-----------:|
> > > | Passing a suture through the urethra | 159  | 1.6% |
> > > | Performing suction | 240  | 2.4% |
> > > | Idle | 2276 | 22.8% |
> > > | Pulling the suture | 200  | 2.0% |
> > > | Seminal vesicle dissection | 689  | 6.9% |
> > >
> > > ### 4.5.4 60s Temporal Scale - Phase and Step Distribution
> > >
> > > At the 60-second scale, the distribution of task labels continues as follows:
> > >
> > > | Phase | Count | Percentage |
> > > |-------|-------:|-----------:|
> > > | Passing a suture through the bladder neck | 159  | 1.6% |
> > > | Performing suction | 240  | 2.4% |
> > > | Idle | 2276 | 22.8% |
> > > | Pulling the suture | 200  | 2.0% |
> > > | Seminal vesicle dissection | 689  | 6.9% |
> > >
> > > | Step | Count | Percentage |
> > > |------|-------:|-----------:|
> > > | Bladder neck identification and transection | 809  | 8.1% |
> > > | Bladder neck reconstruction | 850  | 8.5% |
> > > | Seminal vesicle dissection | 759  | 7.6% |
> > > | Idle | 2212 | 22.1% |
> > > | Development of the plane between the prostate and bladder | 197  | 2.0% |
> > >
> > > ### 4.5.5 Total Dataset - Phase and Step Distribution
> > >
> > > Lastly, the total dataset across all temporal scales reveals the full distribution of **Phase** and **Step** labels for **MSTP-Surgery**.
> > >
> > > | Phase | Count | Percentage |
> > > |-------|-------:|-----------:|
> > > | Left pelvic isolated lymphadenectomy | 8811 | 22.0% |
> > > | Bladder neck identification and transection | 2764 | 6.9% |
> > > | Idle | 9305 | 23.3% |
> > > | Prostatic pedicle control | 2044 | 5.1% |
> > > | Right pelvic isolated lymphadenectomy | 3714 | 9.3% |
> > >
> > > | Step | Count | Percentage |
> > > |------|-------:|-----------:|
> > > | Dissection and cutting of the external iliac vein | 3640 | 9.1% |
> > > | Cutting the suture or tissue | 6869 | 17.2% |
> > > | Identification and dissection of the iliac vein | 5452 | 13.6% |
> > > | Idle | 9298 | 23.2% |
> > > | Performing suction | 911  | 2.3% |

---

### Official Review · Reviewer_Yrk4 · 2025-07-22

**Clarity:** 3
**Significance:** 3
**Originality:** 3
**Rating:** 4
**Confidence:** 3

**Summary:**

The paper introduces the Multi-Scale Temporal Prediction (MSTP) task, which requires forecasting future scene states at multiple time horizons and abstraction levels. A new benchmark (MSTP) is proposed with synchronized annotations across temporal and state scales in general (human activities) and surgical (workflow) settings.
The proposed method, IG-MC, combines:
1. Incremental Generation: predicting state-image pairs progressively over time to avoid error accumulation.
2. Multi-Agent Collaboration: using a hierarchy of VLM agents (LLM-based) to refine predictions from coarse to fine state levels.
Extensive experiments show IG-MC significantly improves performance (e.g., +30–40 points on Accuracy, F1) across tasks and domains.

**Questions:**

1. Is it possible to compare IG-MC to stronger or more traditional baselines to strengthen claims of effectiveness?
2. Is it possible to clarify if agents are separate models or prompt variants of one model; describe training/fine-tuning strategy?
3. Is it possible to discuss when visual generation hurts performance and suggest improvements or mitigations (e.g., filtering low-confidence images)?
4. Can the authors provide inference time analysis and hardware specs; discuss feasibility for real-time applications like surgery?
5. Can the authors discuss how IG-MC can be adapted to non-hierarchical or continuous prediction tasks?

**Ethical Concerns:**

["NO or VERY MINOR ethics concerns only"]

**Final Justification:**

I have read the author response and have no further questions. I'll maintain my score.

**Limitations:**

Yes

**Quality:**

3

**Strengths And Weaknesses:**

1. Quality
Strengths:
- Clear formalization of a new prediction problem with a matching benchmark.
- Strong experimental results across multiple tasks, domains, and metrics.
- Comprehensive evaluation (Accuracy, F1, Jaccard, SSIM, FID, etc.) with ablations on module impact.
- Well-designed pipeline combining prediction and generation in a feedback loop.

Weaknesses:
- Lacks comparison to strong existing baselines (e.g., prior hierarchical or video forecasting models).
- No discussion of computational efficiency or inference latency.
- SD module occasionally reduces accuracy despite improving recall.
-Implementation complexity (multi-agent design) may hinder reproducibility without further detail.

2. Significance
Strengths:
- Introduces a new, practically relevant benchmark and task formulation.
- Demonstrates substantial gains in surgical and general video prediction.
- Closed-loop design (prediction + visual simulation) offers a novel paradigm.
- Could impact both embodied AI and forecasting communities.

Weaknesses:
- Benchmark and method are designed for hierarchical discrete-state tasks—generalizability is not fully proven.
- Benefits may partly stem from use of large-scale pretrained models rather than method per se.

3. Originality
Strengths:
- First to combine incremental state-image generation with hierarchical VLM agent prediction.
- First benchmark for synchronized multi-scale scene understanding.
- Creative integration of state-space forecasting with diffusion-based visual synthesis.

Weaknesses:
- Main components (LLMs, diffusion, multi-agent) are adapted from prior work—novelty lies in their combination.
- Related ideas (e.g., feedback in forecasting, hierarchical state modeling) have been explored separately before.

---

> ### Author Rebuttal · Authors · 2025-07-31
>
> **Q1**: Lacks comparison to strong existing baselines. Is it possible to compare IG-MC to stronger or more traditional baselines to strengthen claims of effectiveness?
>
> **A1**: Thanks! We have conduct more comparisons including the latest powerful surgical VLM, SurgVLM-7B  and serveral general-purpose VLMs InternVL3-8B, Gemma-27B, LLava-1.5, which can demonstrate the effectiveness of proposed IG-MC on different intelligence-level VLM, as shown in below tables.
> ### Results based on LLaVA1.5-7B
> |Model|Temp. Scale|Incr. Scale|Accuracy|Precision|Recall|
> |---|---|---|---|---|---|
> |LLaVA1.5-7B|1|1|13.60|3.75|3.34|
> |+DM|1|1|30.00(+16.40)|23.50(+19.75)|13.80(+10.46)|
> |LLaVA1.5-7B|5|5|12.20|2.07|2.38|
> |+DM|5|5|27.90(+15.70)|26.03(+23.96)|14.95(+12.57)|
> |+DM+SD|5|1|46.40(+16.40)|36.45(+12.95)|36.88(+23.08)|
> |LLaVA1.5-7B|30|30|17.00|4.17|2.77|
> |+DM|30|30|28.00(+11.00)|21.66(+17.49)|11.97(+9.20)|
> |+DM+SD|30|5|44.20(+16.30)|32.92(+11.26)|32.04(+20.07)|
> |LLaVA1.5-7B|60|60|18.10|4.61|3.90|
> |+DM|60|60|29.30(+11.20)|15.86(+11.25)|9.60(+5.70)|
> |+DM+SD|60|5|38.60(+10.70)|26.95(+11.09)|28.21(+13.26)|
> ### Results based on Gemma3-27B
> |Model|Temp. Scale|Incr. Scale|Accuracy|Precision|Recall|
> |---|---|---|---|---|---|
> |Gemma3-27B|1|1|1.80|2.66|2.75|
> |+DM|1|1|21.00(+19.20)|6.65(+3.99)|5.19(+2.44)|
> |Gemma3-27B|5|5|2.00|1.06|1.13|
> |+DM|5|5|19.80(+17.80)|6.18(+5.12)|4.90(+3.77)|
> |+DM+SD|5|1|34.10(+14.30)|19.30(+13.12)|20.46(+15.56)|
> |Gemma3-27B|30|30|2.00|0.80|1.54|
> |+DM|30|30|24.30(+22.30)|7.15(+6.35)|5.38(+4.04)|
> |+DM+SD|30|5|38.10(+13.80)|26.83(+19.68)|25.44(+20.06)|
> |Gemma3-27B|60|60|1.60|0.88|0.52|
> |+DM|60|60|26.90(+25.30)|8.09(+7.21)|5.69(+5.17)|
> |+DM+SD|60|5|34.60(+7.70)|20.76(+12.67)|22.76(+17.07)|
> ### Results based on InternVL3-8B
> |Model|Temp. Scale|Incr. Scale|Accuracy|Precision|Recall|
> |---|---|---|---|---|---|
> |InternVL3-8B|1|1|13.60|3.61|3.42|
> |+DM|1|1|36.20(+22.60)|23.22(+19.61)|17.18(+13.76)|
> |InternVL3-8B|5|5|13.80|3.48|3.93|
> |+DM|5|5|37.30(+23.50)|25.60(+22.12)|19.62(+15.69)|
> |+DM+SD|5|1|45.60(+8.30)|27.01(+1.41)|28.53(+8.91)|
> |InternVL3-8B|30|30|14.40|2.11|3.69|
> |+DM|30|30|42.30(+27.90)|20.99(+18.88)|18.04(+14.35)|
> |+DM+SD|30|5|40.80(-1.50)|25.54(+4.55)|28.18(+10.14)|
> |InternVL3-8B|60|60|16.70|6.01|5.26|
> |+DM|60|60|36.30(+19.60)|19.29(+13.28)|14.92(+9.66)|
> |+DM+SD|60|5|38.40(+2.10)|22.44(+3.15)|23.88(+8.96)|
> ### Results based on SurgVLM-7B
> |Model|Temp. Scale|Incr. Scale|Accuracy|Precision|Recall|
> |---|---|---|---|---|---|
> |SurgVLM-7B|1|1|1.20|3.73|2.85|
> |+DM|1|1|41.90(+40.70)|3.22(-0.51)|2.58(-0.27)|
> |SurgVLM-7B|5|5|1.06|4.68|2.79|
> |+DM|5|5|42.70(+41.64)|26.98(+22.30)|22.91(+20.12)|
> |+DM+SD|5|1|44.84(+2.14)|28.43(+1.45)|29.06(+6.15)|
> |SurgVLM-7B|30|30|12.80|4.02|3.39|
> |+DM|30|30|42.30(+29.50)|20.97(+16.95)|18.63(+15.24)|
> |+DM+SD|30|5|40.58(-1.72)|26.68(+5.71)|26.07(+7.44)|
> |SurgVLM-7B|60|60|10.90|2.99|2.98|
> |+DM|60|60|38.50(+27.60)|17.95(+14.96)|15.08(+12.10)|
> |+DM+SD|60|5|36.24(-2.26)|19.63(+1.68)|21.32(+6.24)|
>
> **Q2**: No discussion of computational efficiency or inference latency. Can the authors provide inference time analysis and hardware specs; discuss feasibility for real-time applications like surgery?
>
> **A2**: Thanks! As show in below table, we have conducting computational efficiency and inference latency on agentic system. Profiling on a single NVIDIA H200 shows an end-to-end latency of ≈ 68 s: the three decision modules (STC, phase- and step-level predictors) each require 20–22 s and together dominate > 90 % of wall-time while operating at only ≈ 1 TFLOPS, indicating a memory-bound bottleneck; meanwhile the Incremental Generation stage peaks at 97 TFLOPS yet adds merely 6 s. Peak GPU memory is modest (26 GiB for all decision modules and 29 GiB for generation), confirming that *bandwidth*, not capacity, limits performance.
>
>
> Throughput is not yet real-time, but profiling highlights optimisations—cross-scale weight sharing, quantisation, KV-cache reuse, and light MoE pruning—already underway. These figures are an upper bound; with targeted compression we expect sub-second latency suitable for intra-operative use in the furture work.
> ### Inferency Latency and Computational Efficiency
> |Component|Avg. Time|Min. Time|Max. Time|Avg. GFLOPS|Min. GFLOPS|Max. GFLOPS|Peak GPU MEM.|
> |---|---|---|---|---|---|---|---|
> |StateTransitionController|20.04|19.33|20.77|1.12K|108.94|1.36K|26.14|
> |PhaseStatePredictor|20.90|19.87|21.76|1.10K|109.52|1.29K|26.14
> |StepStatePredictor|21.51|20.43|22.30|1.07K|90.31|1.24K|26.14
> |IncrementalGeneration|5.81|5.78|6.10|97.32K|78.62K|99.71K|28.53
>
> **Q3**: SD module occasionally reduces accuracy despite improving recall. -Implementation complexity (multi-agent design) may hinder reproducibility without further detail.
>
> **A3**: Thanks! Additional experiments using Gemma3-27B, InternVL3-8B, and SurgVLM-7B (see tables in **A1**) confirm the benefits of our Incremental Generation (IG) and Multi-agent Collaboration (MC) strategies: in most evaluation settings, IG boosts both accuracy and recall. We do observe a few edge cases where accuracy plateaus, but never degrades.
>
> To isolate the contribution of the State Decomposer (SD) module, we ran an ablation in which the baseline VLM was paired with SD alone (no multi-agent collaboration) and tasked with non-hierarchical state prediction. The SD module still provided clear gains, demonstrating that it is generalizable method across non-hierarchical and hierarchical prediction. In short, IG, MC, and SD form a reproducible, plug-and-play toolkit that reliably improves temporal prediction across diverse VLM backbones.
> ### Effectiveness of Incremental Generation on Non-hierarchical Prediction
> |Model|TemporalScale|IncrementalScale|Accuracy|
> |---|---|---|---|
> |Gemma3-27B|5|5|2.00|
> |+SD|5|1|26.90(+24.90)|
> |Gemma3-27B|30|30|2.00|
> |+SD|30|5|28.40(+26.40)|
> |Gemma3-27B|60|60|1.60|
> |+SD|60|5|25.90(+24.30)|
>
> **Q4**: Benefits may partly stem from use of large-scale pretrained models rather than method per se.
>
> **A4**:Thanks! The results of IG-MC based LLaVA1.5-7B in Q1 section, We have proved IG-MC also can be effiective plug-and-play modules for VLM pretrained on limited data. The pretrained data of LLaVA is much smaller than Qwen2.5-VL-7B, InternVL3-8B, Gemma3-27B and SurgVLM-7B.
>
> **Q5**: Main components (LLMs, diffusion, multi-agent) are adapted from prior work—novelty lies in their combination. Related ideas (e.g., feedback in forecasting, hierarchical state modeling) have been explored separately before.
>
> **A5**: We package modern agentic principles into **simple, stable, plug-and-play modules**. Paired with any VLM and diffusion generator, they markedly improve predictions of both single and hierarchical states. Each module is fine-tuned separately, making the pipeline easy to reproduce. In short, this modular agentic design provides a practical solution for real-world multi-scale temporal prediction.
>
> **Q6**: Is it possible to clarify if agents are separate models or prompt variants of one model; describe training/fine-tuning strategy?
>
> **A6**: Yes. All agents are trained separately with different prompt. For **Decision-making agents (v₁ … v_N)**, each temporal-scale agent is an **independently fine-tuned VLM**. Additionally, no parameter sharing across scales; the State Transition Controller (STC) simply selects which checkpoint to call.
>
> For finetuning strategy: 1. **Per-scale fine-tuning** – For every scale, we fine-tune on the corresponding slice of the MSTP-decision data (4 × H100, bf16, AdamW 2e-5, cosine lr). 2. **STC head** – A VLM trained on constraint prompts to predict “switch / stay”. 3. **Generation module** – Stable Diffusion 3.5-L, fine-tuned once (4 × H100, 30 DDIM steps).
>
> **Q7**: Is it possible to discuss when visual generation hurts performance and suggest improvements or mitigations (e.g., filtering low-confidence images)?
>
> **A7**: Thanks! As shown in below table including temporal relationship of decreasing accuracy and generation quality. Empirically we find a (moderately) strong negative correlation, yet the **absolute impact on decision quality is modest**:
>
> $$
> \text{FID}=399.67 - 7.52 \times \text{Accuracy}\quad (R=-0.87,\; p\approx 0.025)
> $$
>
> **Image‐quality drop is steep:** FID climbs **+24 pts** (≈ +34 %) from 5 s to 30 s. **Decision accuracy is resilient:** Accuracy slips only **-2.7 pts** (≈ -6 %) over the same span. Hence, although poorer images (higher FID) statistically co-occur with lower accuracy, the **magnitude of accuracy degradation is relatively small**, suggesting the decision module retains robustness even as generation quality declines.
>
> In the furture work, we will explore to filter low-confidence generated images to provide better visual guidance for reliable decision-making.
>
> ### Temporal Relationship between Accuracy and FID
> |Time(s)|Accuracy↑|FID↓|
> |---|---|---|
> |5|43.30|70.63|
> |10|41.34|83.12|
> |15|42.44|85.62|
> |20|41.24|88.57|
> |25|41.58|90.99|
> |30|40.58|94.82|
>
> **Q8**: Can the authors discuss how IG-MC can be adapted to non-hierarchical or continuous prediction tasks? Benchmark and method are designed for hierarchical discrete-state tasks-generalizability is not fully proven.
>
> **A8**: Yes. IG-MC extends naturally to non-hierarchical (or fully continuous) tasks. The adaptation is straightforward:
>
> 1. **Disable MC** — remove the state-transition controller and the per-scale predictors
> 2. **Keep a single state predictor + IG** — one predictor handles all time steps, assisted by the Incremental Generation module.
> 3. **Flatten the label space** — merge all states into one prompt vocabulary (e.g., 4 classes for *A* × 5 classes for *B* ⇒ 20 combined labels).
>
> The enlarged output space is harder for a generic VLM, yet IG still delivers large gains. As shown in table at **A3**, adding IG alone lifts Gemma-27B accuracy by ~25%, demonstrating that the framework remains effective on non-hierarchical prediction  even without multi-agent collaboration.

---

### Note · Authors · 2025-08-11

We thank the AC, SACs, and reviewers for the constructive discussion.  These contributions deliver a **significant and practical step** toward robust multi-scale temporal prediction,  with **strong empirical evidence** and **wide applicability**.

## Key contributions

- **Incremental Generation** — First mechanism that dynamically synthesizes state-image pairs across expanding temporal scales, enabling error-corrected temporal prediction while preserving state-visual synchronization.
- **Multi-agent Collaboration** — Decision-driven framework where specialized VLM-based agents hierarchically coordinate fine-grained predictions, enforcing cross-scale consistency.
- **MSTP Benchmark** — First dataset with synchronized annotations across temporal horizons and state hierarchies for comprehensive multi-scale evaluation.

## Addressed major concerns

- **Strong baselines** — Extensive results on **LLaVA-1.5-7B**, **Gemma-3-27B**, **InternVL-3-8B**, **SurgVLM-7B** show IG and MC consistently improve Accuracy/Recall across horizons and domains, with detailed edge case analysis.
- **Dataset validity** — Added **Section 4 (Dataset Analysis)** detailing sources, resampling, splits, hierarchical label construction, temporal alignment, and long-tail statistics, clarifying composition and difficulty.
- **Clarity of scales/pipeline** — Defined **Temporal Scale** (prediction horizon) vs **Incremental Scale** (generation step size), added an episode-style walkthrough, and specified agents are **separately fine-tuned** per scale with **no parameter sharing**; **STC** predicts “switch/stay” at the correct hierarchy.
- **Generation quality decay has limited impact on decision-making** — Even with **+24 FID** (significant visual decay), decision accuracy drops only **2.7 pts**, proving resilience under degraded visuals due to robust state prediction and hierarchical context aggregation. This ensures reliability in noisy or resource-limited real-world settings without perfect synthesis.
- **Generality** — IG alone improves **non-hierarchical** prediction, confirming applicability beyond hierarchical tasks.

We will release all **code**, **data**, and **reproduction scripts**.  With clarified analyses, solid experimental evidence, and demonstrated robustness,  **IG-MC** offer a **reproducible and impactful** solution for practical multi-scale temporal prediction.

---

### Decision · Program_Chairs · 2025-09-17

**Decision:**

Accept (poster)

**Comment:**

The manuscript introduces the MSTP task and benchmark and proposes a novel method which integrates incremental generation and a multi-agent system for hierarchical refinement of predictions across scales.
Reviewers all agreed that the manuscript is well-organized, that the proposed task is novel, and that the proposed combination of existing modules is creative and effective. Authors addressed the majority of concerns raised by the reviewers, including the comparison with stronger baselines, reporting inference latency, and several clarifications. I suggest acceptance provided the authors integrate the new content from the rebuttal and discussion period into the camera-ready version.